



# A monthly surface $p$CO$_2$ product for the California Current Large Marine Ecosystem

Jonathan D. Sharp[1,2], Andrea J. Fassbender[2], Brendan R. Carter[1,2], Paige D. Lavin[3,4], Adrienne J. Sutton[2]

[1]Cooperative Institute for Climate, Ocean, and Ecosystem Studies (CICOES), University of Washington, Seattle, WA, 98195, U.S.A.
[2]NOAA/OAR Pacific Marine Environmental Laboratory, Seattle, WA, 98115, U.S.A.
[3]Cooperative Institute for Satellite Earth System Studies/Earth System Science Interdisciplinary Center (CISESS/ESSIC), University of Maryland, College Park, MD, 20740, U.S.A.
[4]NOAA/NESDIS Center for Satellite Applications and Research, College Park, MD, 20740, U.S.A.

*Correspondence to*: Jonathan D. Sharp (jonathan.sharp@noaa.gov)

**Abstract.** To calculate the direction and rate of carbon dioxide gas (CO$_2$) exchange between the ocean and atmosphere, it is critical to know the partial pressure of CO$_2$ in surface seawater ($p$CO$_{2(sw)}$). Over the last decade, a variety of data products of global monthly $p$CO$_{2(sw)}$ have been produced, primarily for the open ocean on 1° latitude by 1° longitude grids. More recently, monthly products of $p$CO$_{2(sw)}$ that are more finely spatially resolved in the coastal ocean have been made available. A remaining

challenge in the development of $p$CO$_{2(sw)}$ products is the robust characterization of seasonal variability, especially in nearshore coastal environments. Here we present a monthly data product of $p$CO$_{2(sw)}$ at 0.25° latitude by 0.25° longitude resolution in the Northeast Pacific Ocean, centered around the California Current System (CCS). The data product (RFR-CCS; Sharp et al., 2021; https://doi.org/10.5281/zenodo.5523389) was created using the most recent (2021) version of the Surface Ocean CO$_2$ Atlas (Bakker et al., 2016) from which $p$CO$_{2(sw)}$ observations were extracted and fit against a variety of satellite- and model-

derived surface variables using a random forest regression (RFR) model. We validate RFR-CCS in multiple ways, including direct comparisons with observations from moored autonomous surface platforms, and find that the data product effectively captures seasonal $p$CO$_{2(sw)}$ cycles at nearshore mooring sites. This result is notable because alternative global products for the coastal ocean do not capture local variability effectively in this region. We briefly review the physical and biological processes — acting across a variety of spatial and temporal scales — that are responsible for the latitudinal and nearshore-to-offshore

$p$CO$_{2(sw)}$ gradients seen in RFR-CCS reconstructions of $p$CO$_{2(sw)}$.

## 1 Introduction

The concentration of carbon dioxide gas (CO$_2$) in Earth's atmosphere has rapidly increased from about 280 parts per million in 1750 to over 400 parts per million today (Joos and Spahni, 2008; Dlugokencky and Tans, 2019). This rise in CO$_2$ concentration is a direct result of human activities such as fossil fuel combustion, deforestation, and agriculture (Ciais et al.,

2014; Friedlingstein et al., 2020). The presence of human-produced or "anthropogenic" CO$_2$ in the atmosphere — along with other anthropogenic greenhouse gases — leads to planetary warming, with a disproportionate amount of heat (~90%) being





absorbed by the ocean (von Schuckmann et al., 2020). A portion of anthropogenic $CO_2$ (~25%) dissolves directly into the ocean (Friedlingstein et al., 2020), mitigating its warming potential. However, dissolved $CO_2$ reacts with seawater to form carbonic acid, which rapidly dissociates and acidifies (primarily) surface ocean environments (Caldeira and Wickett, 2003), with adverse effects for many marine organisms and ecosystems (Orr et al., 2005; Fabry et al., 2008; Pörtner, 2008; Doney et al., 2009; 2020). A primary method for calculating the amount of $CO_2$ transferred to the ocean requires knowing the difference between the partial pressure of $CO_2$ in the atmosphere and surface seawater.

Compared to atmospheric $CO_2$ partial pressure ($pCO_{2(atm)}$), which can be determined with some certainty at a given location even without direct observations due to the well-mixed nature of the atmosphere, surface seawater $CO_2$ partial pressure ($pCO_{2(sw)}$) is more variable and therefore more difficult to constrain (Wanninkhof et al., 2014; Landschützer et al., 2014; Woolf et al., 2019). This variability is a result of ocean mixing, equilibration kinetics between the atmosphere and ocean, biological processes, and thermal effects on $pCO_{2(sw)}$. Filling temporal and spatial data gaps in the observational coverage of $pCO_{2(sw)}$ can therefore be challenging (Hauck et al., 2020; Fay et al., 2021) and a variety of strategies have been attempted over several decades (Takahashi et al., 1993; Rödenbeck et al., 2015), becoming even more prevalent and varied in the literature over time. Briefly, statistical interpolations (Takahashi et al., 1993; 2002; Rödenbeck et al., 2014; Jones et al., 2015; Shutler et al., 2016), multiple linear regressions (Schuster et al., 2013; Iida et al., 2015; Becker et al., 2021), machine-learning-based regression methods (Landschützer et al., 2013; 2014; 2016; 2018; Nakaoka et al., 2013; Zeng et al., 2014; Laruelle et al., 2017; Ritter et al., 2017; Gregor et al., 2017; 2018; Chen et al., 2019; Denvil-Sommer et al., 2019), and biogeochemical-model-based approaches (Valsala and Maksyutov, 2010; Majkut et al., 2014; Verdy and Mazloff, 2017) have been common tactics, each one with its own strengths and weaknesses. Recently, ensemble averages of multiple data- or model-based approaches have become a popular option as well (Gregor et al., 2019; Lebehot et al., 2019; Fay et al., 2021).

One widely used machine-learning-based $pCO_{2(sw)}$ gap-filling strategy relies on a two-step approach consisting of unsupervised clustering using a self-organizing-map (SOM) followed by construction of a feed-forward neural network (FFN) for each cluster (Landschützer et al., 2013). This "SOM-FFN" approach is well-established in the literature (Landschützer et al., 2013; 2014; 2015; 2016; 2018; Laruelle et al., 2017; Ritter et al., 2017; Denvil-Sommer et al., 2019), and is recognized as one of the most effective approaches for filling gaps in the observational $pCO_{2(sw)}$ record (Rödenbeck et al., 2015). The SOM-FFN approach was recently applied to coastal ocean areas, resulting in the first globally continuous, multi-year data product of monthly coastal ocean $pCO_{2(sw)}$ at 0.25° resolution (Laruelle et al., 2017). Even more recently, that coastal product was combined with an updated open-ocean product (Landschützer et al., 2020a) to produce a uniform 12-month climatology of $pCO_{2(sw)}$ across the coastal to open-ocean continuum (Landschützer et al., 2020b; 2020c).

The data products provided by Laruelle et al. (2017) and Landschützer et al. (2020b) — hereafter L17 and L20, respectively — are important advancements toward characterizing $pCO_{2(sw)}$ across the entire ocean domain for carbon budget analyses.





Most data-based estimates of oceanic $CO_2$ uptake have considered only the open ocean (e.g., Rödenbeck et al., 2013; Landschützer et al., 2014; Iida et al., 2015; Denvil-Sommer et al., 2019; Gregor et al., 2019; Watson et al., 2020); however, coastal ocean $CO_2$ uptake is estimated to be about 10% of the open-ocean figure (Laruelle et al., 2010; 2014; Bourgeois et al., 2016; Roobaert et al., 2019; Chau et al., 2021) and may be changing at a different rate relative to open-ocean $CO_2$ uptake

(Laruelle et al., 2018). Therefore, augmenting global open-ocean $pCO_{2(sw)}$ data products to include the coastal ocean is quite valuable (Fay et al., 2021). Despite the greater spatial coverage and temporal resolution offered by these new gap-filled $pCO_{2(sw)}$ data products, significant challenges remain for accurately representing $pCO_{2(sw)}$.

One of those challenges involves characterizing seasonal cycles in $pCO_{2(sw)}$, particularly in the nearshore coastal ocean.

Although the L17 product effectively captures $pCO_{2(sw)}$ seasonality when averaged across relatively large coastal ocean regions, the authors assert that "the coastal SOM-FFN tends to systematically underestimate the amplitude of the seasonal $pCO_2$ cycle" in locations where they can make comparisons with direct observations. This result is logical given that: (1) direct observations are made at discrete locations and times, whereas gridded products are averaged over some spatial area and time, which tempers extremes; and (2) fits obtained via least squares regressions or machine learning methods generally tend to

perform better where temporal and spatial variability is low, and worse where variability is high (Landschützer et al., 2014), such as in the coastal ocean. However, this problem must be addressed if we hope to achieve realistic global representations of $pCO_{2(sw)}$ seasonality, which are necessary for investigating the processes that drive this variability (Roobaert et al., 2019) and for ensuring the fidelity of future air–sea $CO_2$ flux projections (Hauck et al., 2020). Addressing carbon exchange in coastal margins has recently been highlighted as a fundamental and emerging research topic in ocean carbon research (Dai, 2021).


Here, we present a reconstruction of $pCO_{2(sw)}$ (1998–2020) in a broad region of the Northeast Pacific (NEP) that includes the California Current System (CCS), surrounding open-ocean regions, and the highly variable continental shelf of the North American west coast spanning from southern Alaska to Baja California. We apply a random forest regression (RFR) approach (Breiman, 2001) to fill observational gaps, constraining $pCO_{2(sw)}$ across the coastal to open ocean continuum. RFR is less

computationally expensive than fitting a neural network and has been shown to produce results comparable to the SOM-FFN approach in terms of overall performance (Gregor et al., 2017), though the two approaches differ mechanistically and therefore adapt to variability within a training dataset in different ways. We show that the RFR approach in the NEP produces realistic monthly maps of surface $pCO_{2(sw)}$ from 1998 to 2020 and that these maps reliably capture seasonal $pCO_{2(sw)}$ variability in the coastal and open ocean.


We compare $pCO_{2(sw)}$ values from our gap-filled product — "RFR-CCS" — to coastal ocean mooring measurements and other direct observations, and to the available global-scale 0.25° resolution SOM-FFN products in the region (i.e., L17 and L20). We speculate as to why nearshore seasonal cycles are better represented by RFR-CCS than by global-scale gap-filled products, and discuss implications for how to best capture seasonal variability in global products going forward. We describe spatial and





seasonal patterns in $p\text{CO}_{2(sw)}$ revealed by RFR-CCS, and discuss the physical and biological processes that likely produce those patterns. Finally, we compare air–sea $\text{CO}_2$ flux computed from RFR-CCS to that from a recently released $\text{CO}_2$ flux product (Gregor and Fay, 2021), and discuss the implications of sporadic sampling on calculations of $\text{CO}_2$ flux in the coastal ocean.

## 2 Methods

### 2.1 Sea surface $f_{\text{CO}_2}$ data acquisition and conversion to $p\text{CO}_2$

Sea surface $\text{CO}_2$ fugacity ($f_{\text{CO}2(sw)}$) data, along with ancillary variables, were obtained from the Surface Ocean $\text{CO}_2$ Atlas (SOCAT; Pfeil et al., 2013; Bakker et al., 2016) version 2021 (SOCATv2021) for latitudes between 15 °N and 60 °N and longitudes between 105 °W and 140 °W (hereafter referred to as "the study region"). SOCAT is an international effort to synthesize quality-controlled $f_{\text{CO}2(sw)}$ observations for the global surface ocean, and has released datasets of individual surface ocean $f_{\text{CO}2(sw)}$ observations and gridded values since 2011, with annual releases since 2015. SOCATv2021 contains nearly 30.6

million $f_{\text{CO}2(sw)}$ observations.

SOCAT data in the study region were filtered to retain $f_{\text{CO}2(sw)}$ observations with a measurement quality control (QC) flag of 2 ("good") and dataset QC flags of A through D ($f_{\text{CO}2(sw)}$ accuracy of 5 µatm or better). This is identical to the QC procedure followed by the SOCAT team for producing gridded data products (Sabine et al., 2013; Bakker et al., 2016). SOCATv2021

provides ancillary variables along with $f_{\text{CO}2(sw)}$, including contemporaneous observations of sea surface temperature (SST) and sea surface salinity (SSS), as well as atmospheric pressure at the ocean surface ($P_{\text{atm}}$) from the National Centers for Environmental Prediction (NCEP) reanalysis; these values were used only for fugacity to partial pressure conversions (Eq. (1)). Though SST and SSS are considered surface values, it's important to note that these are primarily underway measurements taken a few meters beneath the surface, and that non-trivial differences in temperature and salinity may exist between the

measurement depth and the surface (Robertson and Watson, 1992; Donlon et al., 2002; Goddijn-Murphy et al., 2015; Woolf et al., 2016; Ho and Schanze, 2020; Watson et al., 2020).

Sea surface $\text{CO}_2$ fugacity represents $\text{CO}_2$ partial pressure corrected for the non-ideality of $\text{CO}_2$ gas. It was converted to sea surface $\text{CO}_2$ partial pressure ($p\text{CO}_{2(sw)}$) following (Weiss, 1974):

$$p\text{CO}_{2(sw)} = f_{\text{CO}2(sw)} \cdot exp\left(P_{\text{atm}} \frac{B+2 \cdot \delta}{R \cdot T}\right)^{-1} \tag{1}$$

where $B$ and $\delta$ are virial coefficients, $R$ is the ideal gas constant, and $T$ is SST in Kelvin.

## 2.2 Binning of $p$CO$_{2(sw)}$ observations

Sea surface CO$_2$ partial pressure data were aggregated onto a 0.25° latitude by 0.25° longitude grid for each month from January 1998 to December 2020 using a bin-averaging procedure that consisted of computing the means ($\mu$) and standard deviations ($\sigma$) of all observations of $p$CO$_{2(sw)}$ included within each grid cell. Observations prior to 1998 were excluded as an increase in $f$CO$_{2(sw)}$ data coverage occurs around the start of 1998 and the first full year of SeaWiFS chlorophyll observations (which are used in our procedure to fill gaps in the $p$CO$_{2(sw)}$ dataset) is 1998. For cases in which observations in a given grid

cell originated from two or more platforms (e.g., cruises or autonomous assets), platform-weighted $\mu$ and $\sigma$ were computed by first taking the means and standard deviations of all observations made by each platform, then taking the means of those values. This ensured that all platforms contributing observations to a given grid cell were weighted equally, mitigating unwanted biases toward high-resolution measurement systems (Sabine et al., 2013).

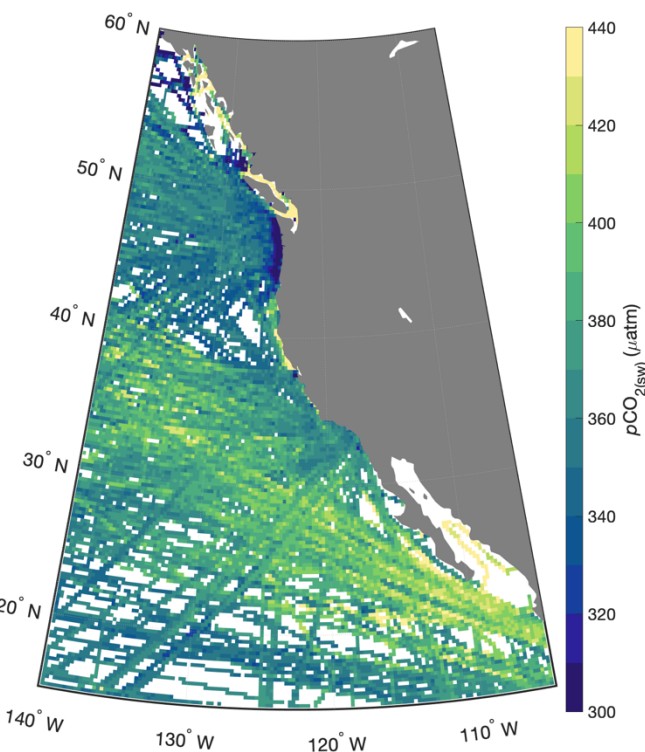


**Figure 1.** Annual mean $p$CO$_{2(sw)}$ from the 0.25° resolution gridded dataset computed as an average over the monthly climatology from 1998 to 2020 for each grid cell. The two extremes of the colorbar can represent $p$CO$_{2(sw)}$ values less than or greater than the colorbar limits; the chosen range represents most of the values and emphasizes regional contrast well.

This bin-averaging procedure is identical to the one followed by the SOCAT team for producing monthly datasets for coastal regions with 0.25° resolution as well as for open-ocean regions with 1° resolution (Sabine et al., 2013; Bakker et al., 2016).





However, here we produced a monthly gridded dataset with 0.25° resolution for a region of the northeast Pacific (15 °N to 60 °N, 105 °W to 140 °W) that spans both the coastal ocean and open ocean. Means of $p\mathrm{CO}_{2(sw)}$ from this gridded dataset (averages over the monthly climatology from 1998 to 2020 for each spatial grid cell) are shown in Fig. 1. Some of the apparent fine-

scale spatial variability in this bin-averaged map is not indicative of true environmental conditions, but originates from the combination of large temporal variability within each grid-cell and uneven sampling of each grid cell across and within years. This form of temporal variability is exactly the kind of spurious result that advanced $p\mathrm{CO}_{2(sw)}$ mapping techniques are intended to circumvent. Figure B1 shows the number of years containing an observation within each month of our gridded $p\mathrm{CO}_{2(sw)}$ dataset. Unsurprisingly, temporal coverage is highest close to the coast, especially in the summer months.


**Table 1.** Sources of data for interpolation of surface $p\mathrm{CO}_2$. Chlorophyll-a (CHL) and mixed layer depth (MLD) were $\log_{10}$-transformed to produce a distribution of values that was closer to normal before constructing the regression model; month of the year was transformed by cosine and sine functions to retain its cyclical nature.

| Predictor Variable | Source | Citation | Original Resolution | | Processing |
| --- | --- | --- | --- | --- | --- |
| | | | Spatial | Temporal | |
| Sea surface temperature (SST) | OISSTv2 | Huang et al. (2021) | 0.25º | daily | monthly averages |
| Sea surface salinity (SSS) | ECCO2 | Menemenlis et al. (2008) | 0.25º | daily | monthly averages |
| Chlorophyll-a (CHL; $\log_{10}$-transformed) | SeaWiFS (1998-2002); MODIS (2003-2020) | NASA Ocean Color | 1/6º | monthly | interpolated to 0.25º resolution |
| Wind speed ($U$) | ERA5 | Hersbach et al. (2020) | 0.25º | monthly | none |
| Atmospheric $p\mathrm{CO}_2$ ($p\mathrm{CO}_{2(atm)}$) | NOAA Marine Boundary Layer Reference $x\mathrm{CO}_2$ | Dlugokencky et al. (2020) | $\sin(lat)$ of 0.05 | weekly | monthly averages, interpolated to 0.25º lat. resolution, multiplied by NCEP sea level pressure |
| Mixed layer depth (MLD; $\log_{10}$-transformed) | HYCOM | Chassignet et al. (2007) | 1/6º | monthly | interpolated to 0.25º resolution |
| Distance from shore | Calculated from gridded lat–lon | Greene et al. (2019) | - | - | - |
| Year | - | - | - | - | - |
| Month of year (converted to two separate predictors using sine and cosine) | - | - | - | - | - |

**2.3 Predictor variable acquisition and processing**

Of the 4,014,844 grid cells that represent the surface ocean gridded in three dimensions at 0.25° resolution over 276 months (1998–2020) in the study region, just 1.25% have an associated gridded $p\mathrm{CO}_{2(sw)}$ value. To fill gaps in this dataset, relationships between $p\mathrm{CO}_{2(sw)}$ and various predictor variables need to be determined. The predictor variables used in this study are primarily



derived from satellite observations or reanalysis models, due to the condition that they be resolved with temporal and spatial
continuity across the study region and time span.

Predictor variables are intended to capture conditions that mechanistically influence $pCO_{2(sw)}$ (e.g., SST and atmospheric $pCO_2$); serve as a proxy for mechanisms that influence $pCO_{2(sw)}$ (e.g., sea surface chlorophyll); or, in the case of temporal and spatial information, constrain additional patterned variability not captured by the mechanistic variables alone. The chosen
predictor variables for this study (Table 1) have all been used before for $pCO_{2(sw)}$ gap-filling methods (e.g., Landschützer et al., 2014; Gregor et al., 2018; Denvil-Sommer et al., 2019; Watson et al., 2020); temporal and spatial predictors were included to ensure robust representation of $pCO_{2(sw)}$ seasonal cycles (Gregor et al., 2017). Included in Table 1 are the sources of each dataset, the original resolutions of each dataset, and the steps that were taken to process each dataset. Appendix A provides more detail about the acquisition and processing of the driver variables, and includes figures showing annual means of selected
variables.

## 2.4 Construction of non-linear relationships using random forest regression

We used the random forest regression approach (Breiman, 2001) to identify relationships between $pCO_{2(sw)}$ and predictor variables in order to fill gaps in the gridded $pCO_{2(sw)}$ dataset. This method averages the results from a number of decision/regression trees (i.e., a "forest") built on bootstrapped replicates of the dataset — which individually have low bias and high variance — to produce a final regression model with reduced variance (Friedman et al., 2001). RFR has not been used extensively for $pCO_{2(sw)}$ gap-filling in the past, though it has been applied more frequently in recent years (e.g., Gregor et al., 2017; 2018; Chen et al., 2019).

Table 2. Model parameters for the random forest regression. Parameter names are the default property names for the MATLAB
TreeBagger class.

| Parameter | Explanation | Value |
|---|---|---|
| NumTrees | Number of decision trees to build for random forest | 1200 |
| MinLeafSize | Minimum number of observations in a given terminal node (i.e., the last node in a decision tree) | 5 |
| NumPredictorsToSample | Number of randomly selected predictor variables to choose from at each node split | 6 |
| InBagFraction | Fraction of input data to sample with replacement for each bootstrapped dataset | 1.0 |

Each decision tree within a random forest regression model is built on a different subset of the training dataset (that contains both the predictor variables and corresponding gridded $pCO_{2(sw)}$ values). This subset is generated by bootstrapping, in which a random set of training data points is selected with replacement — meaning the same data point can be selected more than





once (Breiman, 1996). The number of data points in the bootstrapped dataset is equal to a defined fraction ("InBagFraction" in Table 2) of the original dataset; however, a fraction equal to 1 does not mean the bootstrapped dataset is identical to the original dataset, because selection is made with replacement. Since each regression tree is built on a different subset of the training data, it will contain somewhat different relationships between the predictor variables and the corresponding gridded $pCO_{2(sw)}$ values.


The process of building a decision tree begins at the top "node" of the tree with the values of a single predictor variable being used to split that tree's bootstrapped subset of the training dataset into two smaller subsets (not necessarily of equal size) containing the most similar $pCO_{2(sw)}$ observations. These subsets are then further divided into progressively smaller sets of similar observations, until either the variance among the $pCO_{2(sw)}$ observations in a node drops below a prescribed tolerance

level or the number of observations in the node reaches the user-defined minimum ("MinLeafSize" in Table 2). To ensure that the algorithm does not always pick the same predictor variable (e.g., the one most highly correlated with $pCO_{2(sw)}$ overall) for the split at every node, we limit it to choosing from a different random subset of the predictor variables (equal in number to "NumPredictorsToSample" in Table 2) at each node. This introduces another "random" element into the tree-building process. The random forest contains a large number of these regression trees ("NumTrees" in Table 2) each built on a different, random

bootstrapped subsample of the training data. Once the random forest is built, a set of predictor variables can be provided to the model and the average of the $pCO_{2(sw)}$ values provided by each regression tree is used as the $pCO_{2(sw)}$ prediction for that particular set of inputs.

We constructed a RFR model using the MATLAB "TreeBagger" function with the predictor variables given in Table 1 and

the parameters given in Table 2, along with gridded $pCO_{2(sw)}$ values that were obtained as described in Sects. 2.1 and 2.2. To produce the Northeast Pacific random forest regression $pCO_{2(sw)}$ product (RFR-CCS) that is the main result of this work (Sharp et al., 2021; https://doi.org/10.5281/zenodo.5523389), the full dataset of gridded $pCO_{2(sw)}$ values was used. For optimization and evaluation, subsets of the full dataset were used as described in the following sections.

### 2.5 Optimization of random forest regression model

The predictor variables used (Table 1) and the values for the model parameters (Table 2) were determined by iteratively optimizing the model performance. First, default model parameters were used to train a RFR model using a subset of the data for training (80% of full dataset, distributed randomly across the space and time domains of interest) and a number of possible predictor variables: latitude, longitude, sea surface height, bottom depth, and those given in Table 1. During model selection, the generalization skill for the RFR model was assessed using a validation dataset comprised of 10% of the full dataset, none

of which was included in the training data. After the initial model fit, predictors with a "feature importance" (computed during the RFR fit) significantly lower than all other predictors were sequentially dropped (latitude, longitude, and sea surface height), and this did not substantially change the training or validation root mean squared error (RMSE). Remaining predictor variables





were dropped one at a time for subsequent fits, and the goodness-of-fit and generalization skill of the model were assessed using the RMSE values calculated from applying the model to the training and validation datasets, respectively. The set of predictors with the lowest RMSE after dropping one predictor was carried into the next iteration. If removing a predictor did not increase the validation RMSE significantly, then that predictor was removed from the set of predictors (only bottom depth was dropped in this step). The final set of predictor variables is shown in Table 1.

Next, different values for model parameters (Table 2) were tried iteratively with the retained predictors to identify the optimal values, again by minimizing the RMSE of the validation dataset. Although lower values for the minimum terminal node size performed better in this analysis, additional testing indicated that retaining the default value of 5 was important to prevent overfitting. To determine the appropriate number of trees, we examined how the out-of-bag mean squared error changed as more and more trees were included in the random forest (up to 5000 trees) and selected a number of trees well past the point at which this error had stabilized (1200 trees). Finally, the remaining 10% of the full dataset that was withheld from both the model training and model validation (i.e., the "test data") was used to quantify the mapping uncertainties from the RFR approach (discussed further in Sects. 2.7 and 3.5). The predictor variable feature importances for the final RFR-CCS fit are given in Fig. B2.

### 2.6 Evaluation of random forest regression approach and resulting data product

Once predictor variables and model parameters were optimized, the skill of the RFR approach was further evaluated by splitting the full dataset into different subsets of training data and test data. Evaluation models (RFR-CCS-Evals) were constructed in three different ways: (1) by removing a random (20%) subset of cruises/measurement platforms from the training data (repeated 10 times with different subsets removed each time; $n=10$), (2) by removing all observations from every fifth year from the training data (repeated five times such that data from every year was removed from one of the trials; $n=5$), and (3) by removing all moored autonomous $pCO_{2(sw)}$ measurements (i.e., discrete time series sites primarily located in the coastal ocean) from the training data ($n=1$). The first two strategies were relevant for assessing bulk error statistics for the method applied across the region, and the third strategy for evaluating the ability of the RFR to represent local seasonal variability without the use of high temporal resolution mooring data. These RFR-CCS-Evals are distinct model variants that are only used for assessment; the final RFR-CCS model uses all available training data.

Each data split for an RFR-CCS-Eval was applied directly to SOCATv2021 observations, before bin-averaging the data according to the procedure given in Sect. 2.2; as a result, a gridded training dataset and a gridded test dataset were produced from each split. Data splits were performed in this way to ensure that autocorrelation among measurements from a specific platform did not bias the error statistics. Each split was repeated $n$ times, and error statistics (bias, RMSE, and $R^2$) from comparing $pCO_{2(sw)}$ predicted from the RFR-CCS-Eval models versus $pCO_{2(sw)}$ from the gridded test dataset were averaged.



The final RFR-CCS data product was evaluated through comparisons with gridded $pCO_{2(sw)}$ observations from SOCATv4 (Bakker et al., 2016) and $pCO_{2(sw)}$ observations from surface ocean moorings (Sutton et al., 2019). Of the surface ocean moorings within the study site that are not located within an inland sea and have available data from all twelve months of the year, four (CCE2, NH10, Cape Elizabeth, Châ bá) are located within 40 km of shore and one (CCE1) is about 215 km from

shore. RFR-CCS was also compared to global-scale gap-filled $pCO_{2(sw)}$ products that are available in the region. Namely, we focused on the coastal multi-month $pCO_{2(sw)}$ product from Laruelle et al. (2017; i.e., L17) and the combined coastal and open ocean $pCO_{2(sw)}$ climatology from Landschützer et al. (2020b; i.e., L20).

**2.7 Uncertainty analysis**

Uncertainty in $pCO_{2(sw)}$ for each grid cell was calculated according to the approach used by Landschützer et al. (2014; 2018)

and Roobaert et al. (2019), in which total uncertainty in $pCO_{2(sw)}$ results from a combination of observational uncertainty, mapping uncertainty, and gridding uncertainty. Observational uncertainty ($\theta_{obs}$) is uncertainty inherent to the original measurements of $pCO_{2(sw)}$, evaluated as the average of reported uncertainties in the $f_{CO2(sw)}$ observations from our training dataset, which are flagged by SOCAT with a dataset QC flag of A or B ($f_{CO2(sw)}$ accuracy of 2 μatm or better) and of C or D ($f_{CO2(sw)}$ accuracy of 5 μatm or better); we weighted $\theta_{obs}$ by the number of observations assigned each flag. Mapping uncertainty

($\theta_{map}$) is uncertainty contributed by the RFR mapping procedure, and was evaluated as separate values for the coastal (< 400 km from shore) and open ocean (> 400 km from shore) using the mean of the root mean squared errors for a subset of test data (10%) withheld from both the model training data (80%) and model validation data (10%) (see Sect. 2.5). Gridding uncertainty ($\theta_{grid}$) is uncertainty attributable to aggregating observations into monthly 0.25° resolution grid cells, and was evaluated as separate values for the coastal and open ocean by taking the average unweighted standard deviation among $pCO_{2(sw)}$ values

within each grid cell. These three components were combined to obtain total $pCO_{2(sw)}$ uncertainty ($\theta_{pCO2}$) applicable to each open-ocean grid cell and each coastal grid cell:

$$\theta_{pCO2} = \sqrt{\theta_{obs}^2 + \theta_{map}^2 + \theta_{grid}^2} \tag{2}$$

**2.8 Calculation of $CO_2$ flux**

The flux of $CO_2$ across the ocean–atmosphere interface ($F_{CO2}$) was calculated using a bulk formula:

$$FCO_2 = k_w \times K_0 \times \Delta pCO_2 \tag{3}$$

where $k_w$ is the gas transfer velocity, $K_0$ is the $CO_2$ solubility constant, and $\Delta pCO_2$ is the difference between $CO_2$ partial pressure in seawater and in the overlying atmosphere ($pCO_{2(sw)} - pCO_{2(atm)}$) The salinity- and temperature-dependent equations of Weiss (1974) were used to calculate $K_0$.

Gas transfer velocities were parameterized using a quadratic dependence on wind speed (Wanninkhof, 1992):





$$k_w = \Gamma_{660} \cdot U^2 \cdot \sqrt{660/Sc} \tag{4}$$

where $\Gamma_{660}$ is a gas exchange coefficient normalized to $Sc = 660$, $U^2$ is the squared wind speed, and $Sc$ is the Schmidt number for $CO_2$. Our calculations used $\Gamma_{660} = 0.276$, which is a gas exchange coefficient that is specific to ERA5 reanalysis winds and scaled to a bomb-$^{14}$C flux estimate of 16.5 cm hr$^{-1}$ (Fay et al., 2021). $Sc$ was calculated using the fourth order polynomial fit

of Wanninkhof (2014). $U$ was obtained from ERA5 reanalysis (Hersbach et al., 2020). Flux calculations used monthly averages of squared 3-hourly wind speeds to retain the influence of the quadratic wind term (Fay et al., 2021).

## 3 Results and Discussion

### 3.1 Evaluation by comparison to withheld data

As described in Sect. 2.6, training and test datasets were created by splitting the full dataset prior to bin-averaging. Evaluation

models (RFR-CCS-Evals) were constructed by fitting RFR models using the various gridded training datasets. Values of $pCO_{2(sw)}$ predicted by RFR-CCS-Evals were compared to corresponding values from gridded test datasets. Error statistics (bias, RMSE, and $R^2$) averaged over the $n$ sets of evaluation tests are given in Table 3. When RFR-CCS is compared against all the gridded observations used to construct it, error statistics are predictably strong (last row in Table 3), with a mean bias of 0.00 μatm and an RMSE of 13.33 μatm ($R^2 = 0.93$). These error statistics demonstrate the ability of the RFR model to fit the training

data; the evaluation tests provide insight into the model's ability to predict independent data.

**Table 3.** Error statistics for comparisons of predicted $pCO_{2(sw)}$ from evaluation models versus gridded $pCO_{2(sw)}$ from test datasets. The number of times each test was repeated is given by $n$; where $n$ is greater than one, different subsets of data were removed for each iteration of the test and error statistics are the mean of all iterations.

| Test # | Removed from training dataset | Mean bias (μatm) | RMSE (μatm) | $R^2$ |
|---|---|---|---|---|
| **1 ($n = 10$)** | **Random (20%) subset of cruises** | −0.37 | 26.96 | 0.66 |
| **2 ($n = 5$)** | **Every 5$^{th}$ year** | 1.57 | 30.03 | 0.63 |
| **3 ($n = 1$)** | **Moored autonomous observations** | 8.39 | 43.28 | 0.55 |
| **RFR-CCS** | **None; full dataset used** | 0.00* | 13.33* | 0.93* |

* These statistics represent model training statistics (i.e., evaluated with the same data used to train the model) rather than model validation statistics.

Tests 1 and 2 are good indicators of the overall skill of RFR-CCS. The mean absolute bias for each of those tests is less than 2 μatm and the RMSEs are near or below 30 μatm. These error statistics can be compared with those of L17, who obtained

biases with a mean of 0.0 and RMSEs ranging from 20.5 μatm to 53.1 μatm (mean of 39.2 μatm) for independent evaluations of coastal $pCO_{2(sw)}$ values fit using the SOM-FFN method in ten separate global subregions at 0.25° resolution. For an open-


ocean comparison, Denvil-Sommer et al. (2019) obtained an RMSE of 15.86 for an independent evaluation of $pCO_{2(sw)}$ values fit using a similar neural network approach (LSCE-FFNN) for the subtropical North Pacific (18 to 49 °N) at 1° resolution. The error statistics for our study region, which spans the coastal to open ocean continuum on a finely resolved spatial grid, lie
comfortably between those coastal and open-ocean comparison points.

Test 3 is a good indicator of how well the RFR approach is able to reproduce the values and seasonalities of coastal $pCO_{2(sw)}$ at fixed locations when mooring data at a given location is not provided as training data, as each of the moorings makes continuous $pCO_{2(sw)}$ measurements throughout the year and all but one of the mooring locations included in SOCATv2021 in
this region are within 40 km of shore. The positive mean bias (8.39 µatm) suggests that RFR-CCS somewhat overestimates $pCO_{2(sw)}$ at grid cells corresponding to mooring locations, but this is strongly influenced by high biases at the Cape Elizabeth and Châ bá mooring locations (Table B1). The relatively high RMSE (43.28 µatm) is a result of higher variability in coastal grid cells compared to the open-ocean; this is confirmed by a comparison to the offshore CCE1 mooring (Table B1), where the RMSE from the mooring-excluded RFR-CCS-Eval is just 10.6 µatm.


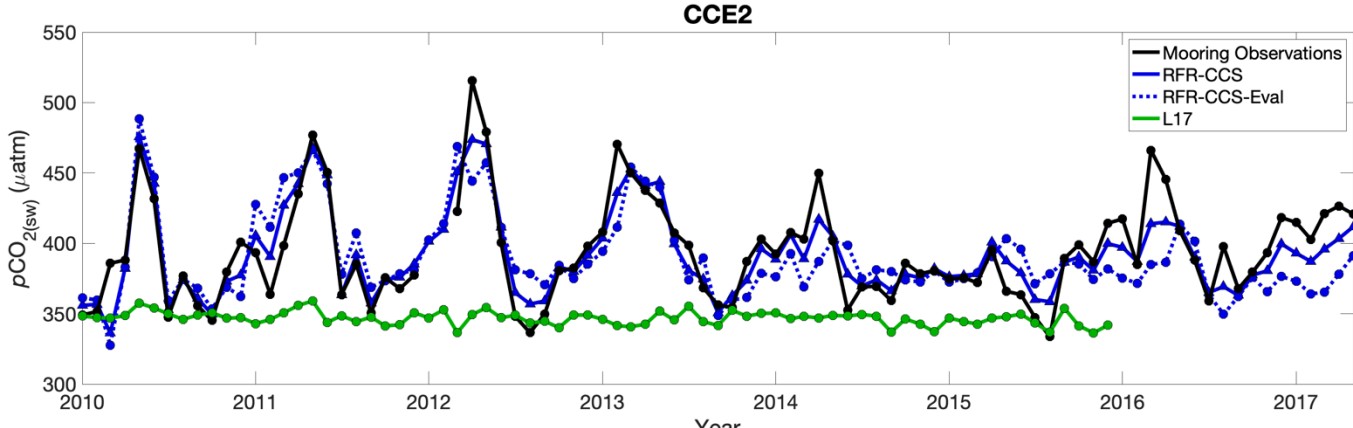

**Figure 2.** Monthly values of $pCO_{2(sw)}$ from mooring observations (black), RFR-CCS (blue solid line), the mooring-excluded RFR-CCS-Eval model (blue dotted line), and L17 (green).

Figure 2 provides an example of one coastal mooring record (CCE2, which is positioned on the shelf break off the coast of Point Conception, CA, at 34.324 °N, 120.814 °W) compared to $pCO_{2(sw)}$ predicted in the corresponding grid cell (centered at 34.375 °N, 120.875 °W) by the mooring-excluded RFR-CCS-Eval model (Test 3) as well as the full RFR-CCS model. For comparison, $pCO_{2(sw)}$ in the same grid cell provided by the L17 coastal product is also shown. At the CCE2 mooring location, RFR-CCS reproduces mooring-observed monthly $pCO_{2(sw)}$ with a mean bias of −2.2 µatm and an RMSE of 16.1 µatm ($R^2$ =
0.81). These error statistics are expected to be relatively favorable, as the RFR-CCS model is trained using mooring observations from CCE2. In contrast, the mooring-excluded RFR-CCS-Eval reproduces monthly mooring-observed $pCO_{2(sw)}$



at CCE2 with a mean bias of −4.6 µatm and an RMSE of 28.9 µatm ($R^2 = 0.41$). This can be compared to the L17 coastal $p$CO$_{2(sw)}$ product, which reproduces monthly mooring-observed $p$CO$_{2(sw)}$ at CCE2 with a mean bias of −44.2 µatm and an RMSE of 57.3 µatm ($R^2 = 0.06$). Notably, the mooring-excluded RFR-CCS-Eval captures $p$CO$_{2(sw)}$ variability at CCE2 more

effectively than the L17 product, even though RFR-CCS-Eval was trained without mooring observations and the L17 training dataset (i.e., SOCATv4) includes CCE2 mooring observations through 2014. Similar results are obtained for comparisons to other mooring records (Table B1; Fig. B3), with RFR-CCS always producing the best error statistics (as expected) and RFR-CCS-Eval always producing a better $R^2$ than L17, indicating that coastal seasonality at mooring locations is better captured by our regional random forest regression model, even when mooring observations themselves are not included in the model

training.

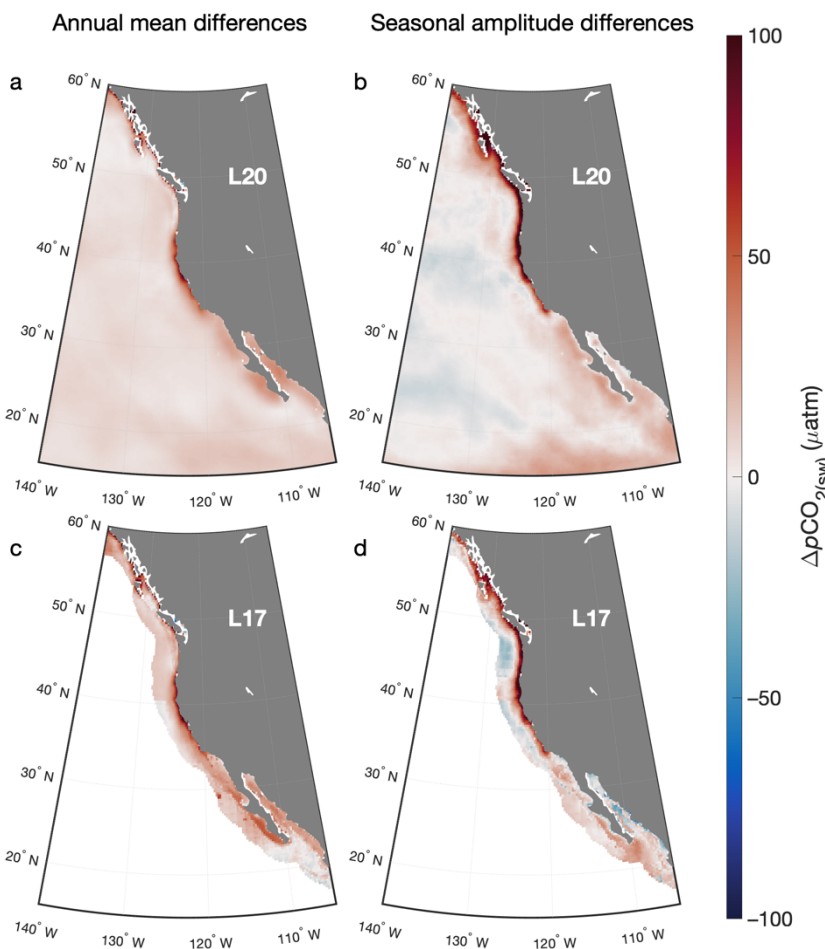

**Figure 3.** Differences between annual means (a, c) and seasonal amplitudes (b, d) of $p$CO$_{2(sw)}$ from RFR-CCS-clim versus the L20 climatology (a, b; RFR-CCS-clim − L20) and versus a climatological average of the L17 product (c, d; RFR-CCS-clim −

L17).





## 3.2 Evaluation by comparison to global $pCO_{2(sw)}$ products

Across the study area, values of $pCO_{2(sw)}$ from RFR-CCS were compared against corresponding values from L17 and L20. For temporal compatibility with L17 and L20, a climatology of average monthly values from RFR-CCS spanning 1998 to 2015 (RFR-CCS-clim) was created for these comparisons. Figure 3 shows mapped differences in annual means and seasonal

amplitudes (calculated as the maximum climatological $pCO_{2(sw)}$ minus the minimum) of $pCO_{2(sw)}$ between RFR-CCS-clim versus L20 (top panels) and RFR-CCS-clim versus a climatological average of L17 (bottom panels); monthly mean differences in are given in Fig. B4.

The most notable feature of the annual mean difference maps is that RFR-CCS-clim produces much higher annual mean

$pCO_{2(sw)}$ than both L17 and L20 in the nearshore coastal ocean and slightly higher $pCO_{2(sw)}$ in the remainder of the study area. Similarly, RFR-CCS-clim produces much higher seasonal variability than both L17 and L20 in the nearshore coastal ocean, especially north of about 34 °N. On average, RFR-CCS-clim produces an area-weighted annual mean $pCO_{2(sw)}$ that is greater than L17 by 19.0 µatm and L20 by 8.4 µatm, and an area-weighted seasonal amplitude that is greater than L17 by 13.0 µatm and L20 by 5.6 µatm.

## 3.3 Evaluation by comparison to gridded observations of $pCO_{2(sw)}$

Values of $pCO_{2(sw)}$ from RFR-CCS, L17, and L20 were compared against the SOCATv4 gridded $pCO_{2(sw)}$ data product. SOCATv4 was used in the development of the coastal L17 product, whereas SOCATv5 was used in the development of the open-ocean product for the merged L20 climatology, and SOCATv2021 was used in the development of RFR-CCS. Therefore, comparisons were made to both the gridded open-ocean observations (1° resolution) and gridded coastal observations (0.25°

resolution) from SOCATv4. To match the resolution of the gridded open-ocean observations from SOCATv4, aggregation from a 0.25° resolution grid to a 1° resolution grid was performed for RFR-CCS, RFR-CCS-clim, and L20. L17 was only compared to gridded coastal observations from SOCATv4, because the two are gridded to the same spatial resolution and cover the same coastally-limited spatial domain.

Figure 4 gives scatter plots of RFR-CCS-clim, L20, RFR-CCS, and L17, each compared against gridded observations from SOCATv4. For comparisons to climatological products (RFR-CCS-clim and L20), gridded SOCATv4 observations were averaged to a monthly climatology across 1998–2015 for consistency with the products. The regional RFR-CCS product and its climatology outperform both global SOM-FFN products: RFR-CCS-clim shows better agreement with gridded monthly means of observations from SOCATv4 than does L20 ($R^2$ = 0.85 [*n=15,212*] versus $R^2$ = 0.73 [*n=15,219*]) and RFR-CCS

(within the coastally-limited spatial domain of L17) shows better agreement with gridded observations from SOCATv4 than does L17 ($R^2$ = 0.96 [*n=8,904*] versus $R^2$ = 0.61 [*n=8,932*]). In particular, the two global products (L20 and L17) struggle to match $pCO_{2(sw)}$ values in the nearshore coastal ocean (within 100 km of the coast), indicated by dark blue symbols in Fig. 4.

Mismatches between global $pCO_{2(sw)}$ products and observations in the nearshore coastal ocean are not unexpected, as regional
error statistics for reconstructed global $pCO_{2(sw)}$ are typically larger than the global mean error statistics (Laruelle et al., 2017;
Landschützer et al., 2020c); and it is generally more challenging to model $pCO_{2(sw)}$ in environments with high temporal and
spatial variability, such as in the nearshore coastal ocean (Landschützer et al., 2014). This result emphasizes the importance of
carefully addressing nearshore $pCO_{2(sw)}$ when constructing global products if one hopes to achieve an accurate representation
of coastal ocean variability. This may be achieved (1) by using a greater number of model clusters for coastal ocean
reconstructions (e.g., L17 uses 10 biogeochemical clusters for the global coastal ocean), (2) by exploring new and different
methods of filling gaps in $pCO_{2(sw)}$ observations (e.g., random forest regression [this study; Gregor et al., 2017; Chen et al.,
2019], gradient boosted decision trees [Gregor and Gruber, 2021], support vector regression [Gregor et al., 2017]), or (3) by
taking an ensemble approach to $pCO_{2(sw)}$ gap-filling (Gregor et al., 2019; Fay et al., 2021).

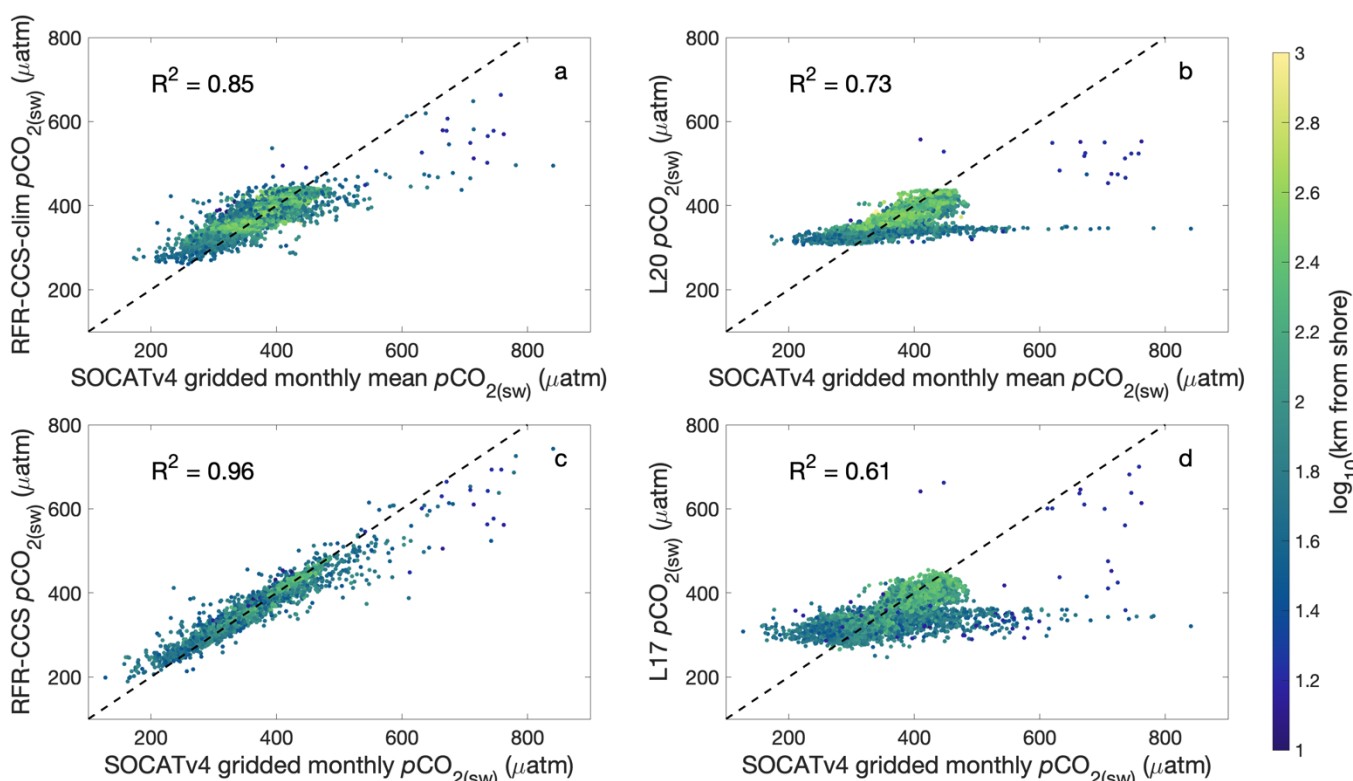

**Figure 4.** Comparisons of $pCO_{2(sw)}$ from (a) RFR-CCS-clim and (b) L20 to SOCATv4 gridded observations that have been
averaged to a climatology; and comparisons of $pCO_{2(sw)}$ from (c) RFR-CCS and (d) L17 to SOCATv4 gridded monthly
observations in the coastally-limited spatial domain of L17. Symbols are color-coded by the base ten logarithm of distance
from shore (km) of each grid cell.


Earth System
Open Access Science Discussions
Data



**Figure 5.** Climatological mean $p\text{CO}_{2(sw)}$ from five NOAA ocean moorings and the corresponding grid cells in RFR-CCS-clim, L20, and a climatological average of L17. Shading represents the standard deviation of all values in each month.




### 3.4 Evaluation by comparison to seasonal observations of $p\text{CO}_{2(sw)}$ at ocean moorings

Values of $p\text{CO}_{2(sw)}$ from RFR-CCS-clim, L17, and L20 were compared against monthly climatologies from mooring observations to evaluate how well each product captured seasonal variability at fixed time-series sites. Figure 5 shows climatologies of mooring-observed $p\text{CO}_{2(sw)}$ (each averaged over available years and normalized to their annual mean) compared to $p\text{CO}_{2(sw)}$ from RFR-CCS-clim, L20, and climatological monthly averages of L17 (each normalized to their annual mean) in the grid cell corresponding to the mooring location. Overall, RFR-CCS-clim does a much better job of capturing the variability in mooring observations than does either L17 or L20 (Table 4).

**Table 4.** Seasonal amplitudes of $p\text{CO}_{2(sw)}$ (µatm) from mooring observations and corresponding grid cells of climatological averages (from 1998–2015) of RFR-CCS-clim, L17, and L20.

| **Mooring:** | **CCE1** | **CCE2** | **Cape Elizabeth** | **Châ bá** | **NH10** |
|:---:|:---:|:---:|:---:|:---:|:---:|
| Mooring | 36.3 | 76.5 | 116.7 | 163.0 | 94.5 |
| RFR-CCS-clim | 32.4 | 64.0 | 133.6 | 129.5 | 97.4 |
| L17 | 21.1 | 6.3 | 37.2 | 35.6 | 26.9 |
| L20 | 23.0 | 6.3 | 22.1 | 21.2 | 18.9 |

### 3.5 Uncertainty calculations

Three components comprised the estimate of uncertainty for $p\text{CO}_{2(sw)}$ values from RFR-CCS: observational uncertainty ($\theta_{obs}$), mapping uncertainty ($\theta_{map}$), and gridding uncertainty ($\theta_{grid}$). According to the procedure detailed in Sect. 2.7, $\theta_{obs}$ was calculated as 3.3 µatm, $\theta_{map}$ as 4.4 µatm for the open ocean and 35.3 µatm for the coastal ocean, and $\theta_{grid}$ as 4.8 µatm for the open ocean and 25.3 µatm for the coastal ocean. These three components were combined to obtain total $p\text{CO}_{2(sw)}$ uncertainty ($\theta_{p\text{CO}2}$) according to Eq. (2), resulting in $\theta_{p\text{CO}2}$ equal to 7.3 µatm for the open ocean and 43.6 µatm for the coastal ocean. The open-ocean value determined through this analysis compares well with the grid-level uncertainty estimated in open-ocean grid cells by Landschützer et al. (2014), which ranged from 8.6 and 17.7 µatm for different regions. The large coastal uncertainty value emphasizes the high degree of variability in monthly $p\text{CO}_{2(sw)}$ near ocean margins.

### 3.6 Spatial and seasonal patterns of sea surface $p\text{CO}_2$

In the open-ocean, relatively high $p\text{CO}_{2(sw)}$ values can be observed off southern Baja California (Fig. 6a) and extending toward the northwest, especially during summer months and into autumn (Fig. 7) when higher sea surface temperatures drive higher $p\text{CO}_{2(sw)}$ (Nakaoka et al., 2013). This area also corresponds to low chlorophyll (Fig. A2) and the lowest wind speeds across





the study region (Fig. A4), suggesting a lack of nutrient delivery and subsequent biological production from deep convection may be producing high $p$CO$_{2(sw)}$ as well. Relatively low open-ocean $p$CO$_{2(sw)}$ values can be observed in the northern part of the study region from about 45 °N to 60 °N (Fig. 6a). Wintertime cooling drives low $p$CO$_{2(sw)}$ in this area, though that effect is compensated by dissolved inorganic carbon (DIC) brought to the surface by deep winter mixing (Ishii et al., 2013). Figure

B5 illustrates competing effects between temperature and winds by displaying correlations between SST and $p$CO$_{2(sw)}$, which are mainly positive below 50 °N, and between wind speed and $p$CO$_{2(sw)}$, which are mainly positive above 50 °N.

In the summer, high biological production in the northern portion of the study region (Fig. A2) removes DIC, keeping $p$CO$_{2(sw)}$ there relatively low. This low-$p$CO$_{2(sw)}$ region extends southward along the California coast to about 34 °N, between both

offshore and nearshore high-$p$CO$_{2(sw)}$ waters. The southward extension of the low-$p$CO$_{2(sw)}$ region is consistent with what we know about the dynamics of the CCS: whereby a narrow band of nearshore waters is high in DIC in the spring and summer due to the direct effects of wind-driven upwelling (Fig. 7), but a wider band of waters farther offshore is lower in DIC due to drawdown by high biological production stimulated by nutrients delivered to the euphotic zone by upwelling (Fassbender et al., 2011; Fiechter et al., 2014; Turi et al., 2014).


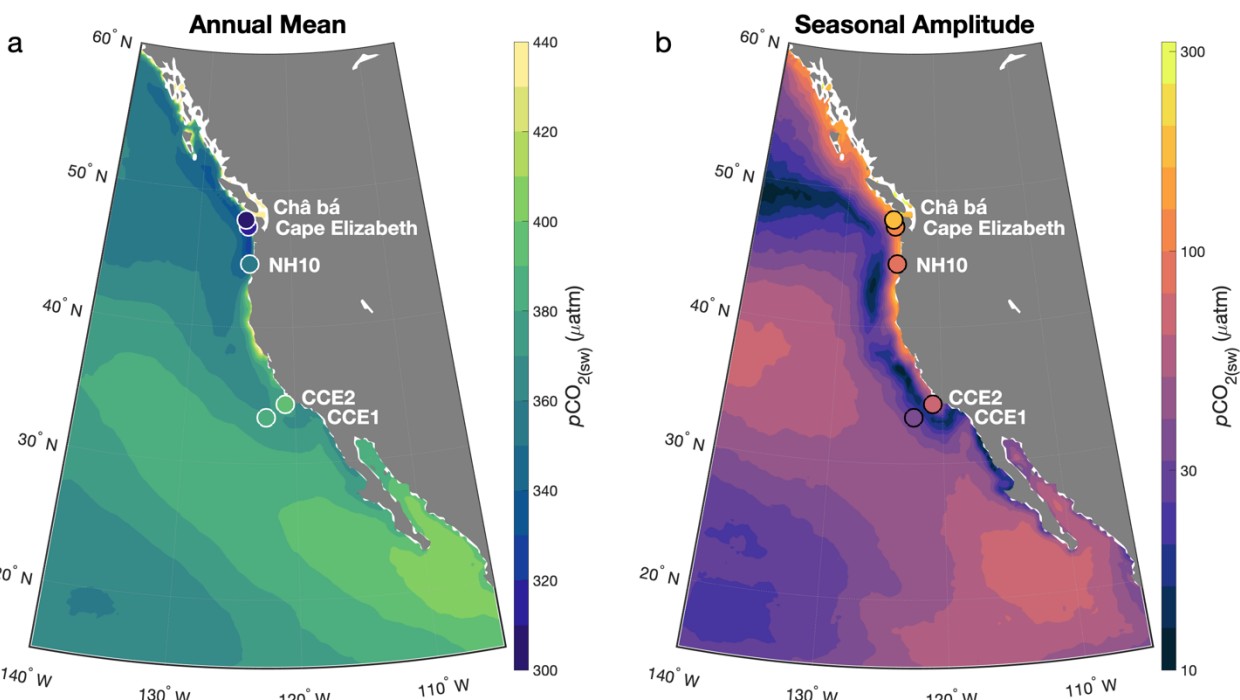

**Figure 6.** Annual mean $p$CO$_{2(sw)}$ (a) and the seasonal amplitude of $p$CO$_{2(sw)}$ (b) from RFR-CCS. Also shown are annual mean $p$CO$_{2(sw)}$ and the seasonal amplitude of $p$CO$_{2(sw)}$ measured at ocean mooring locations.



In the coastal ocean, high $p$CO$_{2(sw)}$ occurs in the central CCS (~34 °N to ~42 °N), with values of 400 µatm or greater beginning in April off Pt. Conception (34 °N) and propagating northward to around Cape Arago (43 °N) through October (Fig. 7). This corresponds to the latitudinal band of the CCS with the strongest and most consistent equatorward winds (Huyer, 1983), which induce upwelling of CO$_2$-rich subsurface waters by wind-driven Ekman transport very near the coast and wind-stress-curl-driven Ekman pumping farther offshore (Checkley and Barth, 2009). This nearshore band of high summertime $p$CO$_{2(sw)}$ has

been previously reported by observational (Hales et al., 2012) and modelling (Fiechter et al., 2014; Turi et al., 2014; Deutsch et al., 2021) studies. It corresponds with naturally low surface pH values and aragonite saturation states, which will be exacerbated by increasing atmospheric CO$_2$ concentrations (Gruber et al., 2012; Hauri et al., 2013), with likely deleterious effects for calcifying organisms (Feely et al., 2008).

Relatively low coastal $p$CO$_{2(sw)}$ values (340 µatm or lower) develop during April off the coasts of Oregon, Washington, and Vancouver Island, and propagate northward toward southern Alaska through September (Fig. 7). Low summertime $p$CO$_{2(sw)}$ in the northern CCS (~42 °N to ~50 °N) has been demonstrated before (Hales et al. 2005; 2012; Evans et al., 2011; Fassbender et al., 2018), and corresponds to the weaker and more variable equatorward winds in summer in the northern CCS (Checkley and Barth, 2009) as well as the effect of DIC drawdown by high primary productivity, which offsets upwelling-induced

increases in $p$CO$_{2(sw)}$. Primary productivity in the northern CCS can be enhanced relative to the rest of the CCS due to factors like riverine nutrient delivery and distribution, submarine canyon-enhanced upwelling, and physical retention of phytoplankton blooms (Hickey and Banas, 2008).

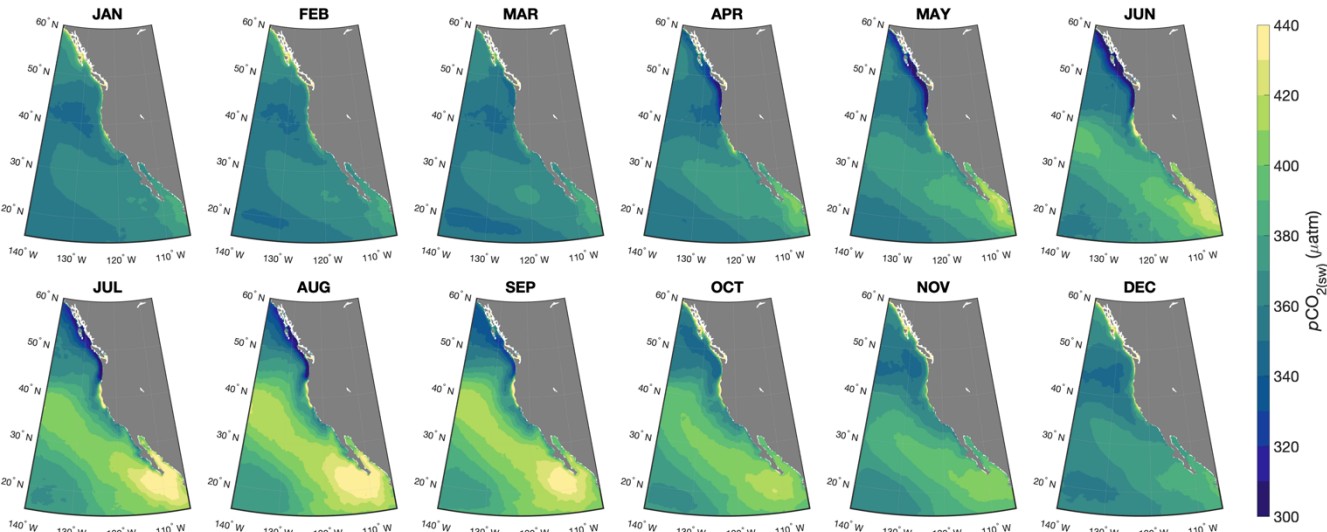

**Figure 7.** Monthly mean $p$CO$_{2(sw)}$ fields from RFR-CCS.

The coastal ocean from Vancouver Island northward is a high-$p$CO$_{2(sw)}$ region from October to March (Fig. 7), which is broadly consistent with observations of high $p$CO$_{2(sw)}$ in the western Canadian coastal ocean during autumn and winter (Evans et al., 2012). This high $p$CO$_{2(sw)}$ is perhaps due to the influence of deep tidal mixing (Tortell et al., 2012) and wintertime light

limitation of DIC drawdown by primary production. The northern coastal area shifts to a low-$p$CO$_{2(sw)}$ region from April to September, again consistent with historical observations (Evans et al., 2012) and likely reflecting surface DIC drawdown by primary production in the region (Ianson et al., 2003).

The coastal ocean from the Southern California Bight (SCB) southward along Baja California (~22 °N to ~34 °N) shows

relatively low $p$CO$_{2(sw)}$ seasonality (Fig. 6b). In this region, $p$CO$_{2(sw)}$ is generally lower than in offshore waters of the same latitude, which matches well with previous results (Fig. 6a; Hales et al., 2012; Deutsch et al., 2021). One exception is directly off the southern tip of Baja California, where especially high summertime $p$CO$_{2(sw)}$ is observed. This may in part reflect the tendency for wind-driven upwelling to bring significant amounts of CO$_2$-rich subsurface waters to the surface just south of major topographic features (van Geen et al., 2000; Friederich et al., 2002; Fiechter et al., 2014). Coastal $p$CO$_{2(sw)}$ within the

Gulf of California (GoC) appears to be strongly influenced by thermally induced seasonal effects, though the lack of observational data coverage in the GoC within SOCATv2021 (Fig. 1), especially within the non-summer months (Fig. B1), may mask more dynamic variability.

The seasonal amplitude of $p$CO$_{2(sw)}$ (Fig. 6b) exhibits interesting variation in the central and northern CCS. Here, nearshore

seasonality is extremely high due to dominant effects from upwelling and primary production; however, seasonality farther offshore is extremely low, likely due to compensating effects by thermally driven changes to $p$CO$_{2(sw)}$ (high temperature in summer increases $p$CO$_{2(sw)}$, low temperature in winter decreases $p$CO$_{2(sw)}$) and biologically/physically driven changes to $p$CO$_{2(sw)}$ (high primary production in summer decreases $p$CO$_{2(sw)}$, deep mixing in winter increases $p$CO$_{2(sw)}$). Elsewhere, a hotspot of high seasonality exists offshore around 40 °N, possibly due to thermal control of $p$CO$_{2(sw)}$ without strong biophysical

compensatory effects.

### 3.7 Carbon uptake in the RFR-CCS domain

A recently published data product (SeaFlux; Gregor and Fay, 2021) described by Fay et al. (2021) harmonizes calculations of global CO$_2$ flux by standardizing the areas covered by different global pCO$_{2(sw)}$ products and by scaling the gas exchange coefficient to different wind products. As part of this procedure, the L20 climatology is used to fill spatial gaps in some of the

pCO$_{2(sw)}$ products. As we have demonstrated here, filling gaps with this climatology may result in an underestimate of the seasonal pCO$_{2(sw)}$ cycle in certain locations, especially nearshore (Fig. 5). For comparison we calculate monthly CO$_2$ flux in our study region from SeaFlux and from RFR-CCS, resulting in the monthly climatologies shown in Fig. 8.



Overall, the SeaFlux ensemble (with ERA5 winds) suggests an oceanic uptake of 69.2 Tg C yr$^{-1}$ for the RFR-CCS domain
between 1998 and 2019 (inclusive) compared to an uptake of 60.0 Tg C yr$^{-1}$ calculated from RFR-CCS. Of the excess 9.2 Tg
C yr$^{-1}$ uptake from SeaFlux, 5.7 Tg C yr$^{-1}$ come from the open ocean and 3.5 Tg C yr$^{-1}$ from the nearshore coastal ocean
(within 100 km of the coast). Given that the nearshore coastal ocean only comprises about 9% of the RFR-CCS region yet
~38% of the discrepancy, the discrepancy in coastal uptake is more significant on a per area basis than the open-ocean
discrepancy, as can be observed visually in Fig. 8. This discrepancy may reflect more coastal outgassing captured by RFR-
CCS than the SeaFlux ensemble, consistent with the annual mean differences shown in Figs. 3a and 3c. Still, the RFR-CCS
results do lie within the variability of SeaFlux pCO$_{2(sw)}$ products (Figs. 8a and 8b).

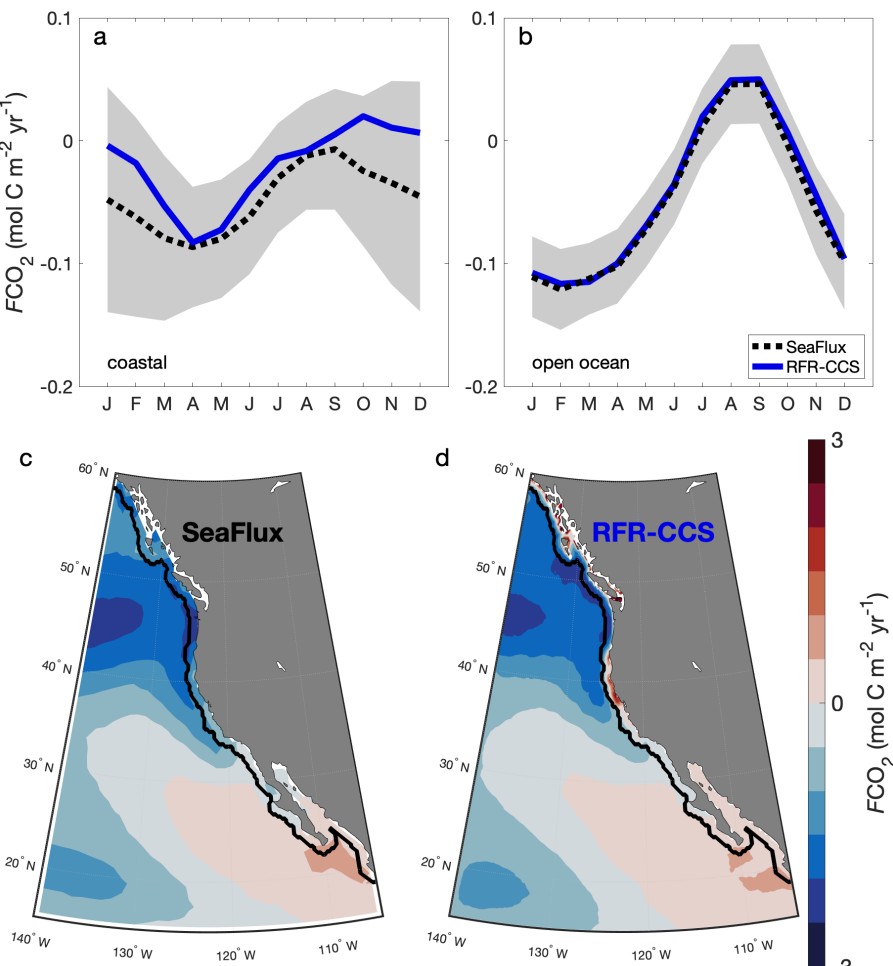

**Figure 8.** Monthly CO$_2$ flux per unit area for the nearshore coastal (a) and open-ocean (b) portions of the RFR-CCS domain,
calculated from SeaFlux (dotted black line) and from RFR-CCS (solid blue line). The grey shaded area represents variability
in SeaFlux pCO$_{2(sw)}$ products, calculated as one standard deviation. Also show is the spatially distributed CO$_2$ flux per unit

area calculated from SeaFlux (c) and from RFR-CCS (d). Red colors indicate net release of $CO_2$ to the atmosphere whereas blue colors indicate net uptake of $CO_2$. The solid black line denotes the boundary between the nearshore coastal (a) and open ocean (b), calculated as 100 km from the coast. All calculations are performed using ERA5 winds and an identical gas exchange coefficient ($\Gamma_{660} = 0.276$).

### 3.8 Effect of sporadic sampling on coastal $CO_2$ flux calculations

RFR-CCS includes $pCO_{2(sw)}$ values for the coastal and offshore ocean in the Northeast Pacific that are representative of monthly conditions. However, air–sea carbon dioxide exchange, which is driven by the difference between oceanic and atmospheric $pCO_2$, operates on shorter timescales. It has been demonstrated in the past that inadequate sampling frequency can be a significant factor biasing $CO_2$ flux ($F_{CO2}$) estimates (e.g., Monteiro et al., 2015).

To demonstrate this potential bias, Fig. 9 shows $F_{CO2}$ at the CCE2 mooring over the course of 2015: (1) calculated from RFR-CCS monthly $pCO_{2(sw)}$ matched to NOAA Marine Boundary Layer monthly $pCO_{2(atm)}$ and monthly averages of squared 3-hourly ERA5 winds (blue), (2) calculated from 3-hourly mooring measurements of $pCO_{2(sw)}$ and $pCO_{2(atm)}$ matched to squared 3-hourly ERA5 winds (grey), and (3) as the one-standard-deviation envelope obtained by the following Monte Carlo process: assigning one randomly selected pair of 3-hourly mooring measurements of $pCO_{2(sw)}$ and $pCO_{2(atm)}$ from each month as the monthly values, matching them with squared 3-hourly ERA5 winds to calculate $F_{CO2}$, and repeating this 100,000 times to obtain statistically meaningful values (green).

The $F_{CO2}$ values provided by the 3-hourly mooring measurements are as close as possible to the true flux. Those provided by RFR-CCS are a best-case scenario for monthly flux approximations in the absence of continuous measurements (because the RFR model was trained on monthly mean $pCO_{2(sw)}$ values from 3-hourly observations at CCE2). Those provided by the Monte Carlo analysis provided reasonable ranges of $F_{CO2}$ that might be obtained from sporadic sampling of one measurement per month without the benefit of an advanced interpolation routine like RFR-CCS.

The annual $F_{CO2}$ calculated from RFR-CCS ($-0.26$ mol C m$^{-2}$ yr$^{-1}$) agrees fairly well with that from the 3-hourly mooring measurements ($-0.18$ mol C m$^{-2}$ yr$^{-1}$). The smaller uptake from the mooring measurements likely reflects the effect of transient outgassing events in the spring and summer, when positive $\Delta pCO_2$ coincides with high wind speeds. The range from the Monte Carlo analysis ($-0.00$ to $-0.36$ mol C m$^{-2}$ yr$^{-1}$) highlights the variety of outcomes in calculated $F_{CO2}$ that might result from sporadic sampling in the coastal ocean, representative of a region with no high-resolution mooring measurements that may be observed by a ship's underway system only a few times a year.



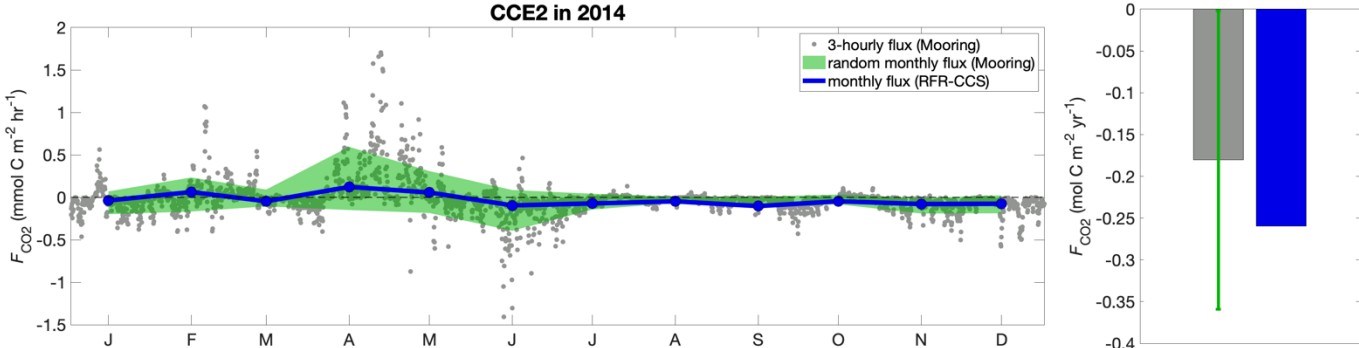

**Figure 9.** Hourly flux of $CO_2$ across the air–sea interface ($F_{CO2}$) calculated from 3-hourly mooring observations (grey), monthly values from RFR-CCS (blue), and the one-standard-deviation envelope of a Monte Carlo analysis ($n$ = 100,000) whereby one randomly-selected 3-hourly mooring observation from each month is selected to represent that month (green). The bar chart on the right gives annual $F_{CO2}$ based on 3-hourly mooring observations (–0.18 mol C m$^{-2}$ yr$^{-1}$) and RFR-CCS (–0.26 mol C m$^{-2}$ yr$^{-1}$), along with the uncertainty in annual $F_{CO2}$ from mooring measurements based on the Monte Carlo analysis (–0.00 to –0.36 mol C m$^{-2}$ yr$^{-1}$).

In large portions of the open ocean, low temporal variability and high spatial correlation means that the aliasing problem may be a relatively low-priority concern for calculations of $F_{CO2}$ from sporadic $pCO_{2(sw)}$ measurements (Bushinsky et al., 2019). However, the dynamic coastal ocean is dominated by processes that influence $pCO_{2(sw)}$ and $F_{CO2}$ on short spatial and temporal scales, making observational frequency a significant factor that can bias annual $F_{CO2}$ calculations. This bolsters the case for the expansion and enhancement of coastal carbon observing systems even with $pCO_{2(sw)}$ gap-filling methods, such as the one described here, at our disposal.

## 4 Conclusions

This work presents a data product, called RFR-CCS, of surface ocean $pCO_2$ in the California Current System and surrounding ocean regions. RFR-CCS was constructed from $pCO_{2(sw)}$ observations in the Surface Ocean $CO_2$ Atlas version 2021 (Bakker et al., 2016), which were related to predictor variables (Table 1) using a random forest regression approach. Validation exercises (Table 3) reveal that this approach is able to predict independent $pCO_{2(sw)}$ values with a skill commensurate to expectations (mean bias near zero and RMSE ≈ 30 µatm), considering the highly variable coastal ocean comprises a large portion of the study region.

RFR-CCS captures variability in $pCO_{2(sw)}$ in the Northeast Pacific, especially at coastal time-series locations, more effectively than do global-scale data products of $pCO_{2(sw)}$. This is evident through comparisons to gridded monthly observations in the SOCAT database and though comparisons to the seasonal amplitudes of $pCO_{2(sw)}$ measured at coastal mooring locations. The





improvements made by RFR-CCS mainly represent the enhanced ability of regional data fits to capture local-scale variability
compared to global data fits. Going forward, perhaps global-scale gap-filled $pCO_{2(sw)}$ products that include a clustering step
would benefit from the creation of a greater number of clusters in the coastal ocean, allowing for more robust reconstruction
of local variability. Improvements detailed here may also be due to the flexibility of RFR in capturing multiple different length
scales of variability (Gregor et al., 2017), which may make the method especially useful for regions that span both the coastal
and open ocean. The CCS is also particularly data-rich, and this work demonstrates the excellent resolution of nearshore
variability that can be achieved in gap-filled $pCO_{2(sw)}$ products when coastal observing systems are sustained over time.

Spatial and seasonal patterns of $pCO_{2(sw)}$ revealed by RFR-CCS reflect interactions of physical and biological processes that
differ substantially with latitude, season, and distance from shore. For example, high annual mean $pCO_{2(sw)}$ in a narrow band
of the central coastal CCS reflects spring and summer upwelling; low annual mean $pCO_{2(sw)}$ and $CO_2$ uptake in the northern
coastal CCS and the offshore CCS in general reflects $CO_2$ drawdown by primary production, largely stimulated by nutrients
delivered by coastal upwelling. Generally, across the study region, interpretations of $pCO_{2(sw)}$ variability and the processes that
drive it coincide with local-scale explanations in the coastal environment, suggesting high heterogeneity in coastal carbon
cycling.

Finally, in the context of sea surface $pCO_2$ gap-filling strategies, this study highlights important factors that should be
considered when working in coastal areas or regions that span the coastal to open ocean continuum. For one, although a global
gap-filled product may demonstrate mean annual values and average seasonal amplitudes of $pCO_{2(sw)}$ that represent a broad
region effectively, this does not mean that local scale variability within that region has been captured just as well. Data-rich
regions like the CCS confirm this notion, especially when variability at consistent time-series sites like moored autonomous
platforms is considered. Misrepresentation of this nature is especially concerning in dynamic nearshore environments, where
local-scale processes can result in surface biogeochemical characteristics that change rapidly over short timescales. These rapid
changes can have direct consequences for local biological responses and for $CO_2$ flux, both of which operate on relatively
short timescales. To address potential errors associated with misrepresentation of $pCO_{2(sw)}$ variability, the spatiotemporal
coverage of carbon observing systems must be improved, especially at ocean margins. Further, innovative implementation and
assessment of machine-learning approaches (Gregor et al., 2017; Gloege et al., 2021), biogeochemical models (DeVries et al.,
2019; Friedlingstein et al., 2020), and ensemble approaches (Lebehot et al., 2019; Fay et al., 2021) should continue to be
explored to best leverage the existing data.





# 5 Appendices

**Appendix A. Processing of predictor variables**

SST (Fig. A1) was obtained from the NOAA daily Optimum Interpolation Sea Surface Temperature (OISST) analysis product (Reynolds et al., 2007; Huang et al., 2021). This data product combines satellite and in situ observations of SST using an optimum interpolation (OI) technique, providing daily SST values at 0.25° resolution. We averaged daily gridded SST values from OISSTv2.1 for each month from 1998 to 2020 to obtain the required monthly 0.25° resolution datasets to match with our

gridded $pCO_2$.

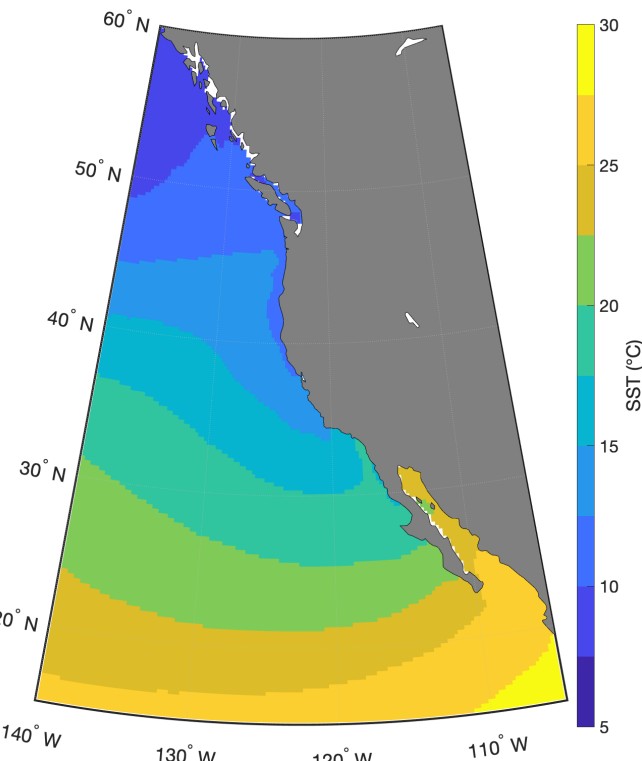

**Figure A1.** Gridded means of SST from satellite observations from 1998–2020.

Sea surface salinity (SSS) was obtained from the NASA Estimating the Circulation and Climate of the Ocean (ECCO) project. The ECCO2 state estimate (Menemenlis et al., 2008) uses a Green's function approach (Menemenlis et al., 2005) to make optimal adjustments to parameters, initial conditions, and boundary conditions of a general circulation model to produce a daily ocean state estimate. We averaged daily gridded SSS values from the ECCO2 state estimate for each month from 1998 to 2020 to obtain the required monthly 0.25° resolution datasets to match with our gridded $pCO_2$.


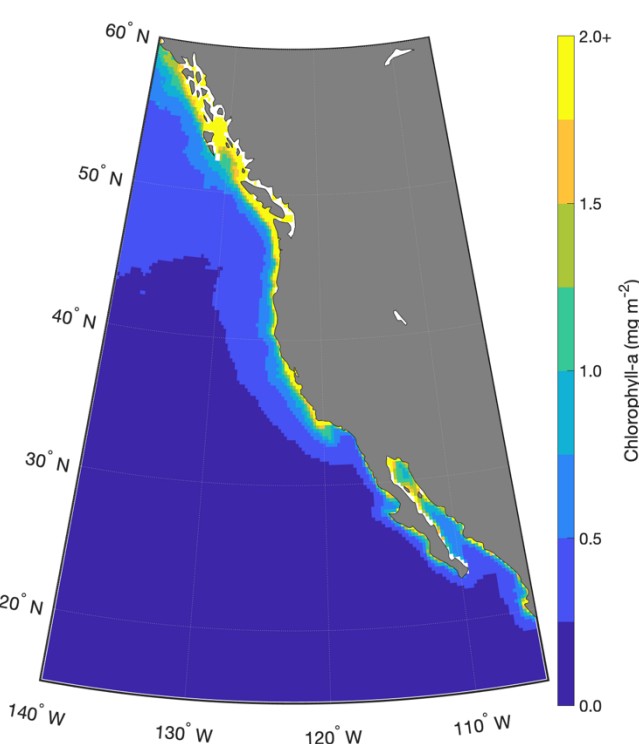

**Figure A2.** Gridded means of Chlorophyll-a concentration from satellite observations from 1998–2020.

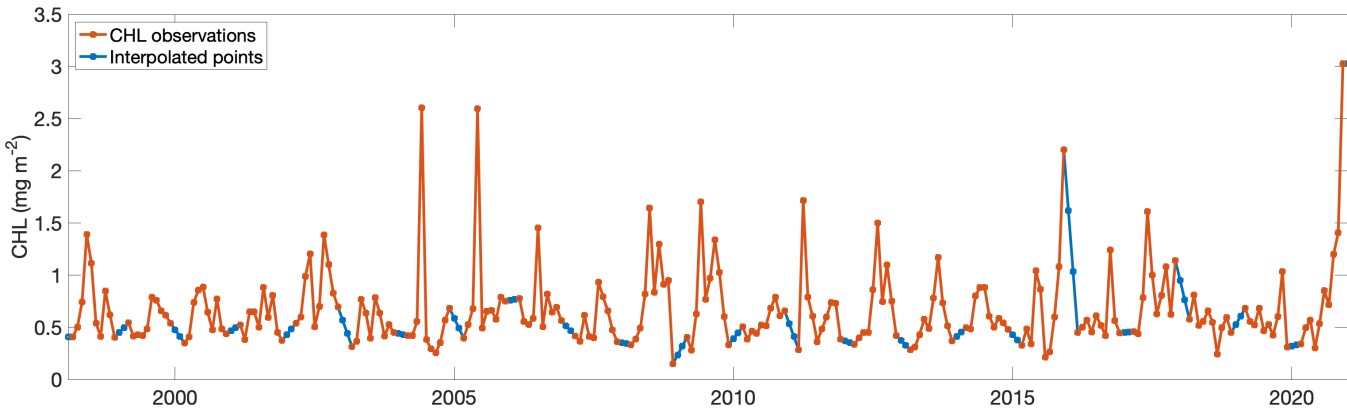

**Figure A3.** Sea surface chlorophyll concentration at 55° N, 135° W. Data from satellite observations are in orange and interpolated data are in blue.

Sea surface chlorophyll-a concentration estimates (CHL; Fig. A2), based on Sea-Viewing Wide Field-of-View Sensor (SeaWiFS) and Moderate Resolution Imaging Spectroradiometer (MODIS) satellite data, were obtained from the Oregon State

University (OSU) Ocean Productivity website (http://www.science.oregonstate.edu/ocean.productivity). The OSU Ocean Productivity website provides both monthly and 8-day CHL files at either 1/6° or 1/12° resolution. We obtained monthly 1/6°

resolution files for 1998–2002 (SeaWiFS-based) and 2003–2020 (MODIS-based), and interpolated each to a 0.25° resolution grid using a standard two-dimensional linear interpolation for each monthly file. For high-latitude wintertime gaps in the CHL datasets, we interpolated CHL for each grid cell through time using one-dimensional linear interpolation when observations in
the previous and subsequent month were available. To avoid anomalous values at the beginning and end of the time series, empty grid cells were filled with nearest-neighbor interpolation when a previous or subsequent observation was not available (Fig. A3). CHL was log$_{10}$-transformed to produce a distribution of values that was closer to normal before constructing the regression model.

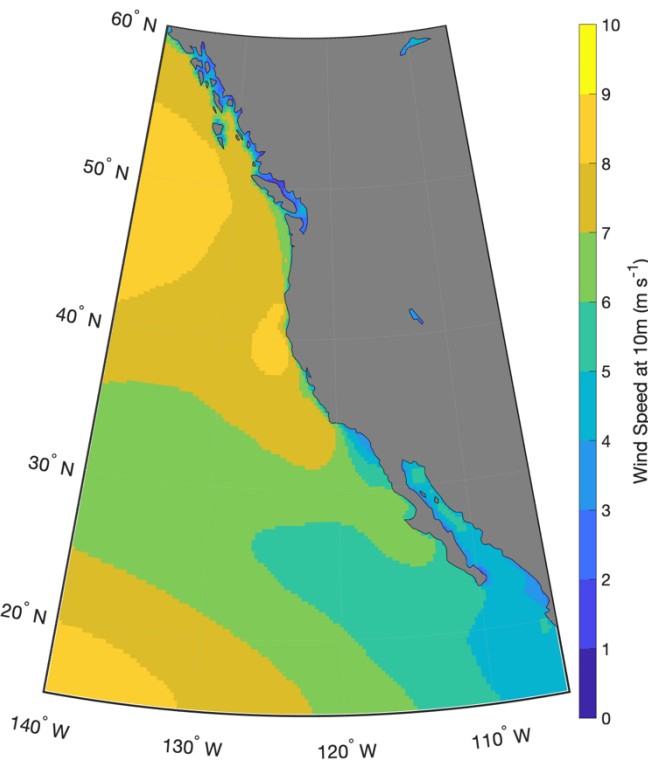


**Figure A4.** Gridded means of wind speed from ERA5 reanalysis from 1998–2020.

Wind speed data (Fig. A4) were obtained from the ERA5 reanalysis product (Hersbach et al., 2020), produced by the European Centre for Medium-Range Weather Forecasts (ECMWF). The ERA5 atmospheric reanalysis provides a detailed record of
atmospheric parameters from 1950 to the present day. We obtained monthly, 0.25° resolution wind speed data at 10 meters above the surface from the Copernicus Climate Change Service (C3S) Climate Data Store (CDS). Wind speed ($U$) was calculated from its vector components (north–south wind, $v_w$, and east–west wind, $u_w$):

$$U = \sqrt{v_w^2 + u_w^2} \tag{A1}$$





Atmospheric $CO_2$ partial pressure ($pCO_{2(atm)}$) was obtained from the NOAA Marine Boundary Layer (MBL) Reference (Dlugokencky et al., 2020). This data product is derived from weekly air samples of atmospheric $CO_2$ mole fraction ($xCO_2$) at a subset of sites from the NOAA Cooperative Global Air Sampling Network. The product is provided as weekly latitudinal averages with a resolution of $\sin(lat) = 0.5$. We interpolated weekly $xCO_2$ values to monthly $xCO_2$ values relative to the middle of each month. To convert $xCO_2$ to $pCO_{2(atm)}$, $xCO_2$ was multiplied by monthly sea level pressure ($P$) from NCEP reanalysis,

which was corrected for water vapor pressure ($VP_{H2O}$) as described by Dickson et al. (2007):

$$pCO_{2(sw)} = xCO_2[P - VP_{H2O}] \tag{A2}$$

Mixed layer depths (MLD), based on output from the Hybrid Coordinate Ocean Model (HYCOM) (Chassignet et al., 2007), were obtained from the OSU Ocean Productivity website. We obtained monthly 1/6° resolution MLD files and interpolated

each to a 0.25° resolution grid using a standard two-dimensional linear interpolation for each monthly file. MLD was $\log_{10}$-transformed to produce a distribution of values that was closer to normal before constructing the regression model.

Distance from shore (*Dist.*) for each grid cell was calculated using the "dist2coast.m" function from the Climate Data Toolbox for MATLAB (Greene et al., 2019), applied to each latitude-longitude grid cell. That function accepts input of latitude and

longitude coordinates and returns the great circle distance to the nearest coastline.

Year (*yr*) was normalized to an epoch of 1997 (i.e., $yr_{norm} = yr - 1997$). Month of year (*mn*) was transformed into two separate predictor variables ($mn_{sin}$ and $mn_{cos}$) using sine and cosine functions to maintain its cyclical nature (after Gregor et al., 2018):

$$mn_{sin} = \sin(2\pi \cdot mn/12) \tag{A3}$$

$$mn_{cos} = \cos(2\pi \cdot mn/12) \tag{A4}$$



## Appendix B. Supplementary figures and tables




**Figure B1.** The number of years containing a $p$CO$_{2(sw)}$ observation within each month over the 23 years of our gridded $p$CO$_{2(sw)}$ data product from 1998–2020.

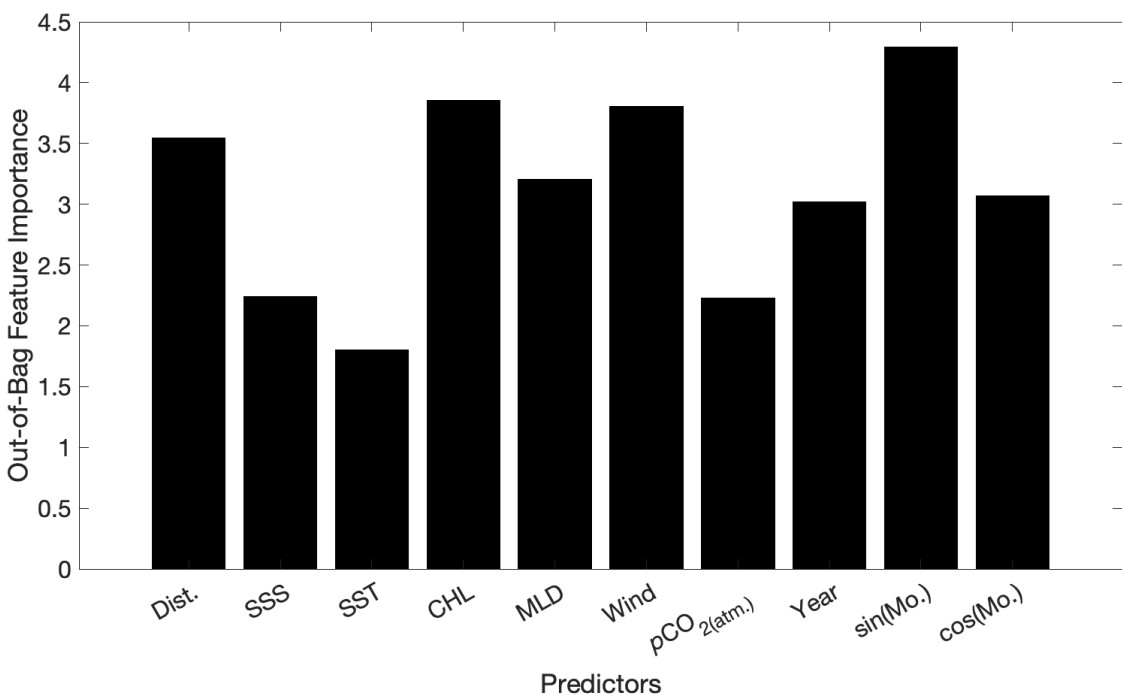


**Figure B2.** Predictor variable feature importances, calculated for the random forest regression model fit used to produce RFR-CCS (Sharp et al., 2021; https://doi.org/10.5281/zenodo.5523389).



**Figure B3.** Like Fig. 2 in the main text, monthly values of $p$CO$_{2(sw)}$ from mooring observations (black), RFR-CCS (blue solid line), the mooring-excluded RFR-CCS-Eval model (blue dotted line), and L17 (green).




**Figure B4.** Monthly mean differences in $p$CO$_{2(sw)}$ values between RFR-CCS-clim and L20 (top) and RFR-CCS-clim and L17 (bottom).




**Figure B5.** Correlations (top) and the p-values of those correlations (bottom) in each grid cell of RFR-CCS between $p\mathrm{CO}_{2(sw)}$ and SST (left) and $p\mathrm{CO}_{2(sw)}$ and wind speed (right).






**Table B1.** Mean biases (MB), root mean squared errors (RMSE), and coefficients of determination ($R^2$) for comparisons of RFR-CCS, the mooring-excluded RFR-CCS-Eval, and L17 to mooring observations.

| Model | CCE1 | | | CCE2 | | | Cape Elizabeth | | | Châ bá | | | NH10 | | |
|---|---|---|---|---|---|---|---|---|---|---|---|---|---|---|---|
| | MB | RMSE | $R^2$ | MB | RMSE | $R^2$ | MB | RMSE | $R^2$ | MB | RMSE | $R^2$ | MB | RMSE | $R^2$ |
| **RFR-CCS** | −1.5 | 8.0 | 0.86 | −2.2 | 16.1 | 0.81 | 8.0 | 24.6 | 0.82 | 12.8 | 29.4 | 0.84 | −7.1 | 26.3 | 0.66 |
| **RFR-CCS-Eval** | −2.0 | 10.5 | 0.77 | −4.6 | 28.9 | 0.41 | 25.8 | 54.8 | 0.47 | 34.9 | 60.5 | 0.48 | −9.2 | 34.3 | 0.49 |
| **L17** | −11.7 | 21.5 | 0.39 | −44.2 | 57.3 | 0.06 | 5.1 | 48.9 | 0.18 | 21.2 | 64.1 | 0.09 | 2.8 | 33.4 | 0.27 |

**6 Code availability**


Matlab code used to process data and create figures included in this manuscript is provided at https://github.com/jonathansharp/RFR-CCS. The majority of this code is also compatible with the open-source software GNU Octave.

**7 Data availability**

The RFR-CCS data product (Sharp et al., 2021) is available as a NetCDF and MATLAB file at https://doi.org/10.5281/zenodo.5523389. Gridded variables used in constructing the product can be accessed at https://figshare.com/articles/dataset/RFR-CCS/15152013.

**8 Author contributions**

JDS, AJF, and BRC contributed to conceptualizing and planning the project. JDS conducted the analysis, produced the data 720 visualizations, and wrote the original draft of the manuscript. JDS, AJF, BRC, PDL, and AJS reviewed and edited the manuscript.

**9 Competing interests**

The authors declare no competing interests

**10 Acknowledgements**

The Surface Ocean $CO_2$ Atlas (SOCAT) is an international effort, endorsed by the International Ocean Carbon Coordination Project (IOCCP), the Surface Ocean Lower Atmosphere Study (SOLAS) and the Integrated Marine Biosphere Research (IMBeR) program, to deliver a uniformly quality-controlled surface ocean $CO_2$ database. The many researchers and funding





agencies responsible for the collection of data and quality control are thanked for their contributions to SOCAT. The moored autonomous $p$CO$_2$ observations are supported by the Global Ocean Monitoring and Observing (GOMO) Program and Ocean Acidification Program of the National Oceanic and Atmospheric Administration. J. Sharp and A. Fassbender were supported by the GOMO Program of the National Oceanic and Atmospheric Administration. P. Lavin was partially supported by NOAA grant NA19NES4320002 (Cooperative Institute for Satellite Earth System Studies – CISESS) at the University of Maryland/ESSIC. This is PMEL Contribution No. 5290. This publication is partially funded by the Cooperative Institute for Climate, Ocean, & Ecosystem Studies (CICOES) under NOAA Cooperative Agreement NA20OAR4320271, Contribution No. 2021-1162.



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
