# Peer review of "A monthly surface *p*CO2 product for the California Current Large Marine Ecosystem"

_Earth System Science Data, 2021_

## Author Comment (AC1)

**Response to Reviewer Comments for** *A monthly surface pCO₂ product for the California Current Large Marine Ecosystem*

The authors thank all three anonymous reviewers for their insightful comments on this manuscript. Below you'll find detailed responses (in bold) to each of the reviewers' comments, which no doubt have improved the clarity of the manuscript and the presentation of our results. A revised version of the manuscript will accompany this document. References to line numbers in this document refer to the revised manuscript, rather than the original submitted version.

Sincerely,
Jonathan D. Sharp and coauthors

**Reviewer 1:**

Sharp and co-authors use a random forest regression approach to map the sea surface pCO2 for the California current system at a high 0.25x0.25 degree resolution. The authors extensively test their approach and compare it with global mapped products from the literature and conclude that their regional approach provides a substantially better representation of the sea surface pCO2, particularly when compared to local measurements.

I believe this is a novel approach and an important analysis and should therefore be published. While it is not particularly surprising that this new local approach outperforms global approaches from L17 and L20, the message is important and clear: Don't rely on global approaches if you are interested in local features.

I am particularly impressed about the effort the authors put into evaluating their approach. Withholding independent observations or full years really guarantees that the observations seen by their machine learning method are truly independent. This should indeed be the standard – well done.

I only have a few minor comments listed below:

1.) I am missing a rationale why random forest regression was chosen? It clearly was a reasonable choice, but I was wondering whether there was a particular motivation to pick this method out of the many available?

**The choice was based largely on early testing of both random forest regression and clustering/neural network fitting. Each method showed similar performance in this region, with slightly better performance by the random forest method (evaluated as with the RMSE of withheld data). Some justification that was previously in the introduction has been moved to section 2.4, and some additional justification was added as well (lines 185–193):**

> **"RFR is the machine learning method of choice for this study as early testing showed better performance than the SOM-FNN method in the Northeast Pacific. Further, RFR**

**is less computationally expensive than fitting a neural network and has been shown to produce results comparable to the SOM-FFN approach in terms of overall performance (Gregor et al., 2017). It should be noted, however, that the two approaches differ mechanistically and therefore adapt to variability within a training dataset in different ways. Finally, while RFR has been explored more frequently in recent years as a method of spatiotemporal $p$CO2$_{(sw)}$ gap-filling both globally (Gregor et al., 2017; 2018) and regionally in the Gulf of Mexico (Chen et al., 2019), far fewer RFR-based $p$CO2$_{(sw)}$ products exist than neural-network-based products. So, this study provides a good opportunity to further demonstrate the utility of RFR for producing monthly fields of $p$CO2$_{(sw)}$, in this case on a regional scale in the Northeast Pacific.”**

2.) Uncertainty: Firstly, I believe the authors overestimate the uncertainty. As shown e.g. by Landschutzer et al 2014, the larger scale (or the full region) error is also dependent on autocorrelation features of the error. Say e.g. your gridding uncertainty for the open ocean is 4.8µatm (based on the std of each grid cell). The standard error for the entire region investigated, however, would scale with the number of uncorrelated grid cells, such that the error E=std/sqrt(N) where N is the number of independent grid cells. For an entirely random (independent) sample of grid cells, N might be very large and the error very small (since grid cell errors cancel each other out). For a dependent (correlated) sample of grid cells, N might be small. So, in a nutshell, by not accounting for the N effect, the authors overestimate their uncertainty. This should be stated

**Clarification has been added to the end of section 2.7 that the uncertainties assessed in this manuscript apply only at the grid cell level, and that uncertainties over larger regions or the entire study area will depend upon the spatial correlation of $p$CO2$_{(sw)}$ values and autocorrelation features of the error: “Whereas $\theta_{p\text{CO2}}$ represents the uncertainty in $p$CO2$_{(sw)}$ for a given grid cell in a given month, uncertainty averaged regionally or over time will not scale exactly with $\theta_{p\text{CO2}}$ due to the spatial correlation of $p$CO2$_{(sw)}$ values and the autocorrelation features of the model error (e.g., Landschützer et al., 2014).” (lines 293–295). This point is reiterated in section 3.5 (lines 461–464).**

3.) Additionally, the authors do a lot of effort calculating the uncertainty, but do not always display it in their figures – Figure 2 and B3 should have an error-shading (corresponding to measurement errors – plus gridding errors for the mapped products) as well, so the reader can see whether differences are within the respective uncertainty.

**Uncertainty shading has been added to Figure 2 and Figure B3. The envelopes around the RFR-CCS estimates represent one standard uncertainty, encompassing the measurement, gridding, and mapping errors evaluated by this manuscript. The envelopes around the L17 estimates represent one RMSE of the independent data evaluation presented by Laruelle et al. (2017) for the spatial cluster associated with the mooring locations. Because measurement errors are relatively low compared to the overall uncertainty for these gridded products, uncertainty envelopes are primarily affected by how well the bin-averaging procedures and regression models are able to represent $p$CO2$_{(sw)}$ for a grid cell, given the influence of natural variability. Therefore, the envelopes around mooring observations equal the standard deviation among all mooring observations within a given**

**month. These error bars provide some context for the magnitude of natural variability. This information is presented in the captions for Figure 2 (lines 345–350) and Figure B3 (lines 735–741).**

4.) Lines 11-12: Not necessarily – other methods (not relying on the pCO2) exist as well

**This opening sentence of the abstract has been changed (lines 11–13). It now reads: "A common strategy for calculating the direction and rate of carbon dioxide gas ($CO_2$) exchange between the ocean and atmosphere relies on knowledge of the partial pressure of $CO_2$ in surface seawater ($pCO_{2(sw)}$), a quantity that is frequently observed by autonomous sensors on ships and moored buoys, albeit with significant spatial and temporal gaps."**

5.) Lines 22-23: "alternative global products" – I agree with the sentence but would remove "alternative". Global products are not an alternative for regional efforts, but target a different research question (e.g. the global ocean CO2 uptake)

**Thanks for pointing out this distinction; the change has been made (line 20).**

6.) Lines 32-33 – It should be noted that the 25% refer to the annual uptake

**This sentence has been edited to clarify that 25% refers to the annual uptake: "About a quarter of annually produced anthropogenic $CO_2$ dissolves directly into the ocean…" (lines 38–39).**

7.) Line 66: The Rodenbeck product actually covers the coastal ocean, but due to its coarse resolution it is not considered a coastal product

**We've altered these few sentences to reflect that coarse resolution in the coastal ocean, in addition to complete exclusion of the coastal ocean, is an issue that must be overcome for data-based estimates of global ocean $CO_2$ uptake (lines 73–77):**

> **"Most data-based estimates of oceanic $CO_2$ uptake have considered only the open ocean (e.g., Landschützer et al., 2014; Iida et al., 2015; Denvil-Sommer et al., 2019; Gregor et al., 2019; Watson et al., 2020) or are based on coarse spatial representations of the coastal ocean (Rödenbeck et al., 2013). However, coastal ocean $CO_2$ uptake is estimated to be about 10% of the open-ocean figure (Laruelle et al., 2010; 2014; Bourgeois et al., 2016; Roobaert et al., 2019; Chau et al., 2021), is far more spatially variable (Liu et al., 2010), and may be changing at a different rate relative to open-ocean $CO_2$ uptake (Laruelle et al., 2018)."**

8.) Line 110: It is more relevant to state how many SOCAT observations exist in the study region

**Thank you for the suggestion. We've added to the end of this sentence "…and over 1.4 million $f_{CO_{2(sw)}}$ observations within the study region" (line 115).**

9.) Line 155 – Table 1: Probably not an issue, but are there any missing chlorophyll values in the study region?

**Yes there are, and this is addressed in Appendix A (lines 680–685). For clarity, we've added to the caption for Table 1 "Gaps in CHL data were filled by linear interpolation over time within each grid cell (see Appendix A)" (lines 150–153).**

10.) Line 198: "most similar pCO2 observations" – what does this mean and how is this determined?

**A point of clarification has been added here: "(i.e., sets of $pCO_{2(sw)}$ observations with the smallest variance among them)" (line 209).**

11.) Line 365 and following: fair point to use SOCATv4 and SOCATv5, but this only partly compensates for the fact, that the RFR-CCS method still was trained with a newer dataset (and the global estimates with observations from the globe)

**We agree with the reviewer that there can be no perfect comparison between a regional data product trained with a subset of data from a given timeframe and a global data product trained with a set of data from an earlier timeframe. For this reason, we chose the SOCATv4 dataset for comparisons, because at the very least this contains data that were available for the construction of all three products (we've added some language indicating that justification in line 394).**

**The results of these comparisons – as well as the issue with them pointed out by the reviewer – emphasize a major point highlighted by this manuscript: that training a gap-filling model based on a large (geographic) subset of data tends to temper regional extremes and produce unrealistic values at the local scale, particularly where variability is high.**

12.) Lines 389-394: I agree with point 1 but disagree with point 2 (for reasons outlined in Gregor et al 2019). Of course local phenomena will only be better represented if a local reconstruction approach is used, but I have my doubt that exploring new methods will overcome this issue – The authors can easily test this by applying their approach to the full coastal domain and then compare the error statistics in the study region.

**While applying this method globally is outside the scope of this research item, we have significantly expanded the discussion at the end of section 3.3 to reflect the reviewer's concerns, particularly as they relate to the work of Gregor et al. (2019). The revised text (lines 414–423) reads as follows:**

> **"This may be achieved (1) by using a greater number of model clusters for coastal ocean reconstructions (L17 uses just 10 biogeochemical clusters for the global coastal ocean), (2) by increasing the spatial and/or temporal resolution of $pCO_{2(sw)}$ data products to better account for small-scale variability (Gregor et al., 2019), (3) by carefully accounting for mismatches between the temperature (and salinity) at which $pCO_{2(sw)}$ is**

**measured versus that at which it is reported in surface data products (Ho and Schanze, 2020; Watson et al., 2020), or (4) by taking an ensemble approach to $pCO_{2(sw)}$ gap-filling to reduce errors overall, and especially in undersampled regions (Gregor et al., 2019; Fay et al., 2021). Ultimately, it will be critical to continue to expand our observational capabilities by means of shipboard underway systems (Pierrot et al., 2009), uncrewed surface vehicles (Meinig et al., 2015, Sutton et al. 2021), biogeochemical Argo floats (Roemmich et al., 2019), moored buoys (Sutton et al., 2019), and other platforms, and to make strides toward incorporating these novel measurements into $pCO_{2(sw)}$ gap-filling schemes (Gregor et al., 2019; Djeutchouang et al., 2022)."**

**Reviewer 2:**

General Comments:

Sharp and coauthors produce a pCO2 observation-based product for the California Current region, expanding and improving upon available coastal products. Their monthly product extends back to 1998 and is at a 0.25 degree spatial resolution and utilizes a Random Forest Regression machine learning technique. Comparisons to other available products as well as in situ observations are thorough and clear.

The paper is clearly written with an important product for the ocean carbon community. I have a few simple questions to clarify and suggestions for improvement of the flow of the text, but other than that support publication of this description of the available product.

Specific Comments:

Overall I feel that the abstract could be improved. For one, it doesn't include the years that this new product (currently) covers. In my opinion, the first half of the abstract reads more like an introductory section and did not give me the information about the excellent product available, the specific improvements it exhibits over other available products, and it's potential uses for the community. I suggest editing to expand on those important aspect further.

**Thank you for these suggestions to improve the abstract. We have indicated the timespan of the data product: "…spanning all months from January 1998 to December 2020." (line 15). We have noted that this regionally-specific product is more representative of CCS conditions than are regional extractions from global products (lines 20–22). Finally, we have added a sentence highlighting some potential uses of the data product for the community: "RFR-CCS will be valuable for investigations of surface carbonate chemistry in the CCS that include the validation of high-resolution models, the attribution of spatiotemporal variability to physical and biological drivers, and the quantification of multi-year trends and interannual variability of ocean acidification." (lines 26–29).**

Also, the sentence that begins on line 36 seems disjointed from the topic of the previous sentences: anthropogenic CO2, impact of dissolving CO2 on ocean life, etc which then jumps to

calculating CO2 transferred between ocean and atmosphere needs the partial pressure. Consider revising to improve flow of this first paragraph.

**We have added a sentence here to bridge the gap between the closing sentence of the paragraph and the preceding topics discussed: "Closing the global carbon budget involves accurately estimating the amount of $CO_2$ taken up by the ocean (e.g., Hauck et al., 2020)." (lines 42–43).**

In the discussion of how the gridded observations is created for this product (Line 135), I would be interested to see how the mean and standard deviation values vary between platforms. Could that be included in the manuscript or is it available through SOCAT somewhere?

**While we don't have any platform-specific diagnostic information to present, we can speculate that the standard deviation in a monthly grid cell comprised of only mooring observations will be far greater than that in a cell comprised of only ship observations. This is due to (1) the locations of moorings in this study area, which are primarily coastal where $pCO_{2(sw)}$ variability is high, and (2) the fact that moorings sample one location throughout the month (at 3-hourly measurement frequency), capturing the full range of monthly $pCO_{2(sw)}$ values, compared to a moving ship which will capture only a snapshot of $pCO_{2(sw)}$ values across a period of a few hours in a given grid cell. More detailed information on the SOCAT gridding procedure is contained in section 3 of Sabine et al. (2013).**

**The gridding uncertainty discussed in our sections 2.7 and 3.5 is determined from the distribution of $pCO_{2(sw)}$ values in monthly grid cells where more than one platform is represented, to provide insight into the uncertainty introduced by bin-averaging over both space and time (i.e. gridding).**

The discussion of your machine learning method is thorough and clear. I find the discussion of three strategies for assessing skill of your model (Section 2.6) very interesting and specifically find the third strategy most exciting as local seasonality is a topic tough to dig into with coastal locations where the seasonality would be enhanced but observations tough to come by. I suggest highlighting the findings related to these tests. I find the results shown in Figure 2 (and associated supplementary figures) to be the most striking and perhaps the most clear take-away from the paper.

**We agree that the insights provided by the third sensitivity test are interesting and exciting. We hope that the discussion in section 3.1 and the indications of these results in the Abstract and Conclusions section make clear that this result is an important take-away. To bolster that notion, we have added another couple sentences at the end of section 3.1 discussing these results and their implications further (lines 367–369):**

> **"This is an important conclusion, especially in light of the recommendation by Hauck et al. (2020) that the inclusion of coastal areas and marginal seas in $pCO_{2(sw)}$ mapping methods will be critical for improving the ocean carbon sink estimate. If these areas are to be included, it is sensible to attempt to capture their unique modes of variability as accurately as possible."**

For Figure 4, is the N value (i.e. the number of dots on each plot) the same and could you include that number on the plot. These types of plots are sometimes hard to glean that information from since so many of the points are overlaid. From its current presentation it doesn't look like the N value would be the same for all 4 subplots.

**The format of Figure 4 was changed to be, rather than a color-coded scatter plot, a two-dimensional histogram color-coded by the average distance from shore of all points that fall into a given box and with transparency set by the relative number of points within each box. Also, the n-value for each comparison is now provided on each plot. This helps with the problem of deciphering among overlain points, and still emphasizes the large nearshore mismatches between L20 and L17 versus the SOCATv4 gridded data.**

For Figure 5, the moorings line has the largest standard deviation/shading. I was wondering if that uncertainty bound is also calculated from the monthly mean values, as would be the case for the products being shown as comparisons (for L20, L17, and RFR- CCS-clim). I wanted to ensure it wasn't simply that difference that is causing the larger spread in the moorings observations vs the products.

**We thank the reviewer for pointing this out. The uncertainty envelope around the mooring data in Figure 5 was indeed a standard deviation among all individual mooring measurements within a given month, rather than a standard deviation among monthly mean values, as is represented by the uncertainty envelopes around the products. The figure has been adjusted accordingly.**

Section 3.5 has a clear discussion of uncertainty calculations and the three main sources of uncertainty considered by the authors. With a stated uncertainty of 43.6uatm in the coastal regions for this product, how do the authors suggest users go forward with that information and utilize this product given that large uncertainty range? Also, this ties into the Figure 8a plot where no uncertainty range is provided for the RFR-CCS seasonal cycle. I would imagine that given an uncertainty that large, that the spread on the resulting flux estimates would be significant. I would suggest changing the plot to include the individual estimates from the products included in the SeaFlux product (rather than the shading showing the range) and then include a shading for the uncertainty associated with the RFR- CCS product. Obviously, each of the products included in SeaFlux would have their own associated uncertainties, similar to those described here for RFR-CCS, but would be good to see regardless. Overall though, I find the comparisons discussed in Section 3.7 to be strong, clear, and decisive.

**The stated uncertainty in the coastal region for this product is based on the combination of measurement, mapping, and gridding errors at the grid cell level (section 2.7). It is not equivalent to the uncertainty in $pCO_{2(sw)}$ values averaged regionally or temporally, as those quantities would scale with the autocorrelation features of the error (now clarified in lines 293–295 in response to a comment from Reviewer 1).**

**Because of difficulties associated with estimating the autocorrelation characteristics of grid-cell-level uncertainties across our study site, we've chosen not to display an**

uncertainty envelope around the climatological flux values in Figure 8a and 8b. We have, however, added individual ensemble members from SeaFlux. We've chosen to retain the one standard deviation shading representing the distribution among those six SeaFlux ensemble members calculated using ERA5 winds.

Overall/Next step Comments:

I finished the paper wishing that there was more discussion on how to extrapolate this work to all coastal regions, or at least eastern upwelling regions with similar physics. Is there "sufficient" data anywhere else in the coastal regions to do such a product? Would the authors expect that the relationships from the RFR would be similar in other coastal regimes? Or would other machine learning methods perhaps be better utilized in areas with less data density? Your conclusions does touch on this somewhat, suggesting that the neural network approach use an increased number of clusters for the coastal ocean areas, but I wonder if you could comment more on this. Could SeaFlux be updated with this (and other) regional coastal products to help fill in missing areas omitted by the open ocean products? Or will the open-ocean products gradually start expanding their coverage to coastal regions with improved machine learning techniques catered specifically to coastal as you have done here?

**Regarding the extrapolation of this method to other coastal ocean regions, we now speculate certain areas that may contain sufficient data for such an analysis to be employed, while noting that the RFR relationships are likely to differ between ocean regions with distinct physical and biogeochemical settings (lines 623–627). Discussion has also been added about other steps (besides increasing the number of coastal clusters in a SOM-FFN approach) that could be taken to improve the fidelity of coastal $pCO_{2(sw)}$ reconstructions (lines 414–423).**

**Other questions posed here we think are more difficult to answer. Gaps in SeaFlux ensemble members that are currently filled with the Landschützer et al. (2020b) climatology could be filled by this and other regionally-specific data products instead, but the application of that idea would certainly pose challenges. Modifying the procedure to construct open-ocean products to function differently in the coastal ocean seems like a good path, as pioneered by Laruelle et al. (2017). Whatever path, it is our hope that future iterations of oceanic $pCO_2$ products do begin to consider the coastal ocean more carefully, especially as the surface observing network grows and more data become available for model training and evaluation.**

**Reviewer 3:**

general comments:

The manuscript applies a machine-learning technique to a surface pCO2 data set in order to fill the gaps of the data set. The resulting data product is assessed at multiple levels, against field data and similar data products: its uncertainties are determined, its vulnerabilities are discussed,

it is compared to other data products, its features are described, as well as its effect on the flux calculations.

Overall, the article is of very high quality. The subject is current and of high importance in our current state of the climate. It is the result of a great amount of work, well thought out and very clearly presented.

The methodology is very sound. the authors properly selected, processed and referenced their data. The machine learning method is clearly described. Their evaluation of the model selected is thorough and proper, as detailed above.

The visualization of the data and the product is sufficient and clear for the most part (see one specific comment).

I also think that the separation of material between the main text and the appendices is appropriate.

Lastly, the manuscript is well written, clear and concise and does a good job at explaining complex concepts.

I only have a few minor comments, I leave it up to the authors whether they want to take the few suggestions below into consideration.

**We thank the reviewer for the kind words and positive review.**

specific comments:

- On line 117, the authors mention that they back calculated the pCO2 from fCO2 values. I am wondering why since the correction is small and would not affect the air-sea difference?

**We chose to present our data product in terms of $p$CO$_2$ rather than $f$CO$_2$ primarily for consistency with previously released data products, many of which also chose to present gridded values of surface $p$CO$_2$ (e.g., Landschützer et al. 2020a, 2020b; Laruelle et al., 2017; Rödenbeck et al., 2013; Gregor and Fay, 2021).**

- around line 120, the authors mention the fact that SST and SSS are not really surface values, due to intake depths. At the same time, I think it is also important to note that these data have not been QC'd.

**An additional sentence has been added to this paragraph to reflect that SST and SSS are not assigned their own QC flags by SOCAT: "Also, while SST and SSS are not assigned explicit QC flags in SOCAT, these parameters do undergo quality control checks during the calculation of $f_{CO2(sw)}$ (Lauvset et al., 2018)." (lines 126–127)**

- The authors assess the effect of data availability on the model (section 2.6) and the effect of sporadic sampling on the coastal flux estimates. It would have been interesting to assess the

sensitivity of the model to entire cruises, which would give a realistic idea of cruise data dependence.

**Though we don't meticulously compare the predictions made by models with certain cruises omitted from the training data to models with those cruises included in the training data, we do provide that analysis in a bulk sense with Test #1 (Table 3). That sensitivity test was conducted by fitting ten individual evaluations models, each with 20% of the available platforms (cruises or moorings) omitted from the training data. Error statistics were then determined by comparing the predicted $pCO_{2(sw)}$ values to the SOCAT values. So, comparison between the last line in Table 3 (error statistics when all data are included in the model training) to the first line (error statistics for the described Test #1) can give some insight into this question. Still, it is worthwhile to note that differences will be magnified compared to any given open-ocean cruise due to the inclusion of highly variable coastal mooring and coastal cruise measurements in this test.**

- In figure 8, the comparison of the SeaFlux and RFR-CCS products would be better served by plotting the difference between the 2. As I said, just a suggestion.

**We agree with the reviewer's suggestion that displaying the difference between the two products would be informative. An additional panel has been added to Figure 8 showing the difference between the RFR-CCS flux values and those from SeaFlux.**

technical corrections:

- line 516: replace "Also show.." by "Also shown...'

**Thanks, this correction has been made.**

**References (not including the quotes from the manuscript included in this document)**

Gregor, L., Lebehot, A. D., Kok, S., and Monteiro, P. M. S. A comparative assessment of the uncertainties of global surface ocean $CO_2$ estimates using a machine-learning ensemble (CSIR-ML6 version 2019a) – Have we hit the wall? Geosci. Model Dev., 12, 5113–5136. https://doi.org/10.5194/gmd-12-5113-2019, 2019.

Gregor, L. and Fay, A. SeaFlux: harmonised sea-air $CO_2$ fluxes from surface $pCO_2$ data products using a standardised approach (2021.04.03) [Data set]. Zenodo. https://doi.org/10.5281/zenodo.5148795, 2021.

Hauck, J., Zeising, M., Le Quéré, C., Gruber, N., Bakker, D. C. E., Bopp, L., Chau, T. T. T., Gürses, Ö., Ilyina, T., Landschützer, P., Lenton, A., Resplandy, L., Rödenbeck, C., Schwinger, J., and Séférian, R. Consistency and Challenges in the Ocean Carbon Sink Estimate for the Global Carbon Budget. Front. Mar. Sci., 7, 1–22, https://doi.org/10.3389/fmars.2020.571720, 2020.

Landschützer, P., Gruber, N., Bakker, D. C. E., and Schuster, U. Recent variability of the global ocean carbon sink. Global Biogeochem. Cycles, 28, 927–949, https://doi.org/10.1002/2014GB004853, 2014.

Landschützer, P., Gruber, N., Bakker, D. C. E. An observation-based global monthly gridded sea surface $pCO_2$ product from 1982 onward and its monthly climatology (NCEI Accession 0160558). Version 5.5. NOAA National Centers for Environmental Information. Dataset. https://doi.org/10.7289/V5Z899N6, 2020a.

Landschützer, P., Laruelle, G., Roobaert, A., Regnier, P. A combined global ocean $pCO_2$ climatology combining open ocean and coastal areas (NCEI Accession 0209633). NOAA National Centers for Environmental Information. Dataset. https://doi.org/10.25921/qb25-f418, 2020b.

Laruelle, G. G., Landschützer, P., Gruber, N., Ti, J. L., Delille, B., and Regnier, P. Global high-resolution monthly $pCO_2$ climatology for the coastal ocean derived from neural network interpolation. Biogeosciences, 14, 4545–4561. https://doi.org/10.5194/bg-14-4545-2017, 2017.

Rödenbeck, C., Keeling, R. F., Bakker, D. C. E., Metzl, N., Olsen, A., Sabine, C., and Heimann, M. Global surface-ocean $pCO_2$ and sea-air $CO_2$ flux variability from an observation-driven ocean mixed-layer scheme. Ocean Sci., 9, 193–216. https://doi.org/10.5194/os-9-193-2013, 2013.

Sabine, C. L., Hankin, S., Koyuk, H., Bakker, D. C. E., Pfeil, B., Olsen, A., Metzl, N., Kozyr, A., Fassbender, A., Manke, A., Malczyk, J., Akl, J., Alin, S. R., Bellerby, R. G. J., Borges, A., Boutin, J., Brown, P. J., Cai, W.-J., Chavez, F. P., Chen, A., Cosca, C., Feely, R. A., González-Dávila, M., Goyet, C., Hardman-Mountford, N., Heinze, C., Hoppema,M., Hunt, C.W., Hydes, D., Ishii, M., Johannessen, T., Key, R. M., Körtzinger, A., Landschützer, P., Lauvset, S. K., Lefèvre, N., Lenton, A., Lourantou, A., Merlivat, L., Midorikawa, T., Mintrop, L., Miyazaki, C., Murata, A., Nakadate, A., Nakano, Y., Nakaoka, S., Nojiri, Y., Omar, A. M., Padin, X. A., Park, G.-H., Pater- son, K., Perez, F. F., Pierrot, D., Poisson, A., Ríos, A. F., Salisbury, J., Santana-Casiano, J. M., Sarma, V. V. S. S., Schlitzer, R., Schneider, B., Schuster, U., Sieger, R., Skjelvan, I., Steinhoff, T., Suzuki, T., Takahashi, T., Tedesco, K., Telszewski, M., Thomas, H., Tilbrook, B., Vandemark, D., Veness, T., Watson, A. J., Weiss, R., Wong, C. S., and Yoshikawa-Inoue, H.: Surface Ocean $CO_2$ Atlas (SOCAT) gridded data products, Earth Syst. Sci. Data, 5, 145–153, https://doi.org/10.5194/essd-5-145-2013, 2013.